# Neural Tangent Kernel Perspective on Parameter-Space Symmetries

## Abstract

Parameter-space symmetries are transformations that modify model parameters without altering the outputs. These transformations can be leveraged to accelerate optimization and enhance generalization. Remarkably, applying a single transformation either before or during training often suffices to realize these benefits. While the effectiveness of this approach is very promising, its underlying mechanisms remain poorly understood. In this paper, we offer an explanation within the Neural Tangent Kernel (NTK) framework, by analyzing how such transformations affect the kernel's properties. In particular, we show that maximizing the alignment between the loss gradient and the data kernel is equivalent to maximizing the alignment between the NTK and the data. Since kernel alignment is known to correlate with optimization rate in the NTK limit, this insight elucidates how loss gradient optimization facilitates faster training. To establish the validity of this approach, we prove that parameter-space symmetries preserve the NTK limit.

## 1 Introduction

*Parameter-space symmetries* are transformations that modify a neural network's parameters while preserving its outputs. Formally, let $F$ be a hypothesis function parameterized by $\theta \in \mathbb{R}^N$, mapping an input space $X \subseteq \mathbb{R}^{d_x}$ to an output space $Y \subseteq \mathbb{R}^{d_y}$, i.e.,

$$F(\theta) : X \to Y . \tag{1}$$

A parameter-space symmetry is a transformation $\theta \mapsto g(\theta) = \theta'$, such that:

$$F(\theta) = F(\theta') . \tag{2}$$

Parameter-space symmetries are common in neural networks, and are useful both in applications and in the analyses of neural network performance. Continuous symmetries, for example, have been identified in many network architectures, including networks with homogeneous activations Badrinarayanan et al. (2015), ReLU networks Neyshabur et al. (2015), radial rescaling activations Ganev et al. (2021), and components such as softmax and batch normalization Kunin et al. (2020).

Parameter permutation symmetries have been linked to the structure of minima in optimization landscapes Simsek et al. (2021); Entezari et al. (2021); Hecht-Nielsen (1990); Chen et al. (1993). Quiver representation theory provides a unified framework for analyzing symmetries in neural networks with pointwise Armenta & Jodoin (2021) and rescaling activations Ganev & Walters (2022). Additionally, Zhao et al. (2022b) identify a class of nonlinear, data-dependent symmetries. Parameter-space symmetries have also proven useful in the analysis of mode connectivity Ainsworth et al. (2022).

An important application of parameter-space symmetries is their potential to accelerate the optimization of neural networks Kunin et al. (2020); Van Laarhoven (2017); Grigsby et al. (2022). One approach, known as *neural teleportation*, achieves this effect by inserting teleportation steps during gradient descent, where the norm of the empirical loss gradient is maximized over the training data via symmetry transformations Armenta et al. (2023); Zhao et al. (2022a). Although this approach reduces the number of gradient descent steps needed to minimize the loss, it is not as efficient if many teleportation steps are needed, as maximizing the gradient norm is computationally expensive. However, Zhao et al. (2023) have demonstrated that even a single gradient loss norm maximization transformation prior to training can meaningfully improve the optimization rate.

While the theoretical foundations of how neural teleportation can improve optimization have been studied Zhao et al. (2022a; 2023), the mechanisms by which a single symmetry transformation can accelerate long-term optimization remain poorly understood.

One principal approach to studying such complex questions in neural networks is to examine them in the large-width limit. In this regime, known as the *neural tangent kernel* (NTK) limit, networks behave linearly with respect to their parameters, and their training dynamics are greatly simplified Jacot et al. (2018); Lee et al. (2019); Li et al. (2019); Yang (2019b; 2020); Yang & Littwin (2021); Shem-Ur & Oz (2024); Chizat et al. (2019); Liu et al. (2020). Despite its simplicity, this first-order approximation captures many of the core properties of realistic finite networks, making it a powerful tool for analyzing their behavior Li et al. (2019); Yang & Hu (2020); Littwin et al. (2021).

The primary objective of this paper is to elucidate the effects of a single parameter-space symmetry transformation through the lens of the neural tangent kernel limit.

**Main Contributions**

- We extend the results of Liu et al. (2020) on the role of the Hessian spectral norm in the linearization of wide neural networks, by explicitly analyzing its effect on the linearization in the function space (Section 3.4).

- We prove that the NTK limit convergence rate is preserved under parameter-space symmetry transformations (Section 4).

- We introduce a new definition of NTK alignment, and show that it serves as an effective indicator of the optimization rate in the NTK regime (Section 5, Appendix G).

- We show that maximizing the norm of the loss gradient once, an established technique for improving optimization, is equivalent to maximizing the kernel alignment (Section 5).

- We present a method to extend any hypothesis function such that any linear invertible transformation, not necessarily a symmetry of the original function, can be realized as a symmetry (Appendix D).

## 2 BACKGROUND

### 2.1 AN EXAMPLE OF PARAMETER-SPACE SYMMETRIES IN FULLY CONNECTED NEURAL NETWORKS

To illustrate the concept of parameter-space symmetries, we consider the fundamental example of fully connected neural networks with a homogeneous activation function.

Fully connected neural networks are characterized by $L + 1 \in \mathbb{N}$ parameter vectors, the biases, denoted by $\theta^{(1)}, \ldots, \theta^{(L+1)}$, $L + 1$ parameter matrices, the weights, denoted by $\theta^{(1,0)}, \ldots, \theta^{(L+1,L)}$, and an activation function $\phi : \mathbb{R} \to \mathbb{R}$. Given an input $x \in X$, the network's layers are recursively defined as follows:

$$F^{(1)} = \theta^{(1,0)} x + \theta^{(1)},$$
$$\forall l = 1, \ldots, L : F^{(l+1)} = \theta^{(l+1,l)} \phi\left(F^{(l)}\right) + \theta^{(l+1)}, \tag{3}$$

where the network's output is given by the final layer:

$$F(\theta)(x) = F^{(L+1)}(\theta)(x). \tag{4}$$

The depth of the network is defined as the number of hidden layers, $L \in \mathbb{N}$, while the width of each hidden layer $l = 1, \ldots, L$ is the size of the layer $n_l \in \mathbb{N}$. The widths of the output and input layers are denoted by $n_{L+1} \in \mathbb{N}$ and $n_0 \in \mathbb{N}$, respectively.

Assuming a homogeneous activation function $\phi$ of degree $d$, i.e., for all $r, c \in \mathbb{R}$, $\phi(cr) = c^d \phi(r)$, then for any inner layer $l = 1, \ldots, L$ and neuron index $i_l = 1, \ldots, n_l$, the following transformation constitutes a symmetry, as defined in Equation 2, for any $c_{i_l} \neq 0$:

$$\theta^{(l+1,l)}_{\cdot, i_l} \mapsto \frac{1}{c_{i_l}^d} \theta^{(l+1,l)}_{\cdot, i_l}, \quad \theta^{(l,l-1)}_{i_l, \cdot} \mapsto c_{i_l} \theta^{(l,l-1)}_{i_l, \cdot}, \quad \theta^{(l)}_{i_l} \mapsto c_{i_l} \theta^{(l)}_{i_l}. \tag{5}$$

Moreover, any combination of such transformations constitutes a symmetry as well. This result extends to activation functions that are homogeneous over a subspace of coefficients, with $c_{i_l}$ restricted accordingly. For instance, in the case of ReLU activation, any $c_{i_l} > 0$ is permissible.

## 2.2 THE NEURAL TANGENT KERNEL LIMIT

In this section, we review the main analytical tool used in this work, the Neural Tangent Kernel limit. To simplify notation and reduce clutter, we omit explicit summation in the equations. In Appendix B, we expand all summations explicitly, and present all indices for clarity.

### 2.2.1 DEFINITION

Analyzing neural networks is notoriously challenging due to their highly non-linear architectures, and vast parameter spaces. Questions such as the one posed in the introduction, namely, how a single parameter-space symmetry transformation can improve optimization, are exceedingly difficult to address analytically.

A principal approach to tackling such questions is to study neural networks in the large-width limit. In this regime, wide networks trained via gradient descent-based algorithms, evolve as though they were linear in their parameters Jacot et al. (2018); Lee et al. (2019); Li et al. (2019). Specifically, the network function can be approximated by its first-order Taylor expansion around the initialization Lee et al. (2019); Yang (2019a); Hanin & Nica (2019); Seleznova & Kutyniok (2022a;b); Lee et al. (2022); Huang & Yau (2020):

$$F(\theta) = F(\theta_0) + \nabla F(\theta_0)^T (\theta - \theta_0) + O\left(\frac{1}{\sqrt{n}}\right) . \tag{6}$$

Here, $F$ denotes the network function, $\theta$ the vector of parameters, $\nabla F$ the Jacobian with respect to $\theta$, and $\theta_0$ the initialization. The $O(1/\sqrt{n})$ term is in the stochastic big-$O$ sense, where $n$ denotes the network's width, i.e., the size of the smallest hidden layer.

Formally, let $F^{\text{lin}}$ denote the linear approximation of $F$ around $\theta_0$, as given in Equation 6. Suppose both $F$ and $F^{\text{lin}}$ are trained using the same gradient descent-based algorithm, under the setup outlined in Yang & Littwin (2021). Then, for any fixed training step $s \in \mathbb{N}$, and input $x \in X$, the difference between the outputs of $F$ and $F^{\text{lin}}$ is asymptotically bounded by $O(1/\sqrt{n})$:

$$\left\| F(\theta(s))(x) - F^{\text{lin}}(s)(x) \right\| = O\left(\frac{1}{\sqrt{n}}\right) , \tag{7}$$

where $\theta(s)$ is $\theta$ at step $s$, and $F^{\text{lin}}(s)$ is $F^{\text{lin}}$ for the $s$-th learning step.

This result holds across a broad class of wide neural network architectures, and gradient-based optimization methods Yang (2019b; 2020); Yang & Littwin (2021); Chizat et al. (2019); Shem-Ur & Oz (2024); Liu et al. (2020).

### 2.2.2 PROPERTIES

A fundamental result of the neural tangent kernel limit is that the network's evolution during training becomes decoupled from its parameters $\theta$. Instead, it is governed by a fixed kernel, defined at initialization. This kernel is a two-point matrix function of size $n_{L+1} \times n_{L+1}$:

$$\forall x, a \in X : \quad \Theta(x, a) = \eta \nabla F(\theta_0)(x)^T \nabla F(\theta_0)(a) , \tag{8}$$

where $\eta > 0$ is the learning rate, and the inner product is taken over all network parameters.

This behavior is well illustrated in the setting of deterministic gradient descent; Let $X' \subseteq X$ be a finite set of inputs, with corresponding outputs $y(X')$, where the network is trained by minimizing the empirical loss $\mathcal{L}$, derived from a loss function $\mathcal{C}$, such as:

$$\Delta \theta = -\eta \nabla \mathcal{L}(\theta) , \tag{9}$$

with the loss defined as:

$$\mathcal{L}(\theta) = \frac{1}{|X'|} \sum_{x \in X'} \mathcal{C}(F(\theta)(x), y(x)) . \tag{10}$$

In the NTK regime, the output evolution follows a semi-linear equation of motion that is decoupled from $\theta$, once the kernel and initial state are chosen:

$$\forall x \in X : \quad \Delta F(\theta)(x) = F(\theta + \Delta\theta)(x) - F(\theta)(x) =$$
$$-\frac{1}{|X'|} \Theta(x, X') \mathcal{C}'\big(F(\theta)(X'), y(X')\big) + O\left(\frac{1}{\sqrt{n}}\right) , \tag{11}$$

where $\mathcal{C}'$ denotes the derivative of the loss with respect to the network output. This derivative is a vector indexed by $i = 1, \ldots, n_{L+1}$ and $x \in X'$, which we also denote by $f$:

$$f_i(\theta)(x) = \mathcal{C}_i'(F(\theta)(x), y(x)) . \tag{12}$$

Likewise, $\Theta(x, X')$ is a matrix indexed by $i$ and $(j, x')$ for $i, j = 1, \ldots, n_{L+1}$ and $x' \in X'$. The product in Equation 11 is standard matrix-vector multiplication. Detailed indexing conventions are provided in Appendix B.

A particularly simple case arises with the mean squared error loss, $\mathcal{C}(y, F) = \frac{1}{2}(F - y)^2$. Here, the loss derivative is just the prediction error:

$$f = \mathcal{C}'(F, y) = F - y , \tag{13}$$

and the dynamics reduce to:

$$\Delta f = -\frac{1}{|X'|} \Theta(\cdot, X') f(X') + O\left(\frac{1}{\sqrt{n}}\right) . \tag{14}$$

Despite being a simplification, this first-order approximation captures many of the key properties of finite-width neural networks, which makes it a valuable approach to understanding finite neural networks Li et al. (2019); Yang & Hu (2020); Littwin et al. (2021).

## 3 PRELIMINARIES

### 3.1 OVERVIEW

The aim of this paper is to explain the result of Zhao et al. (2023), where it was shown that choosing a parameter-space symmetry that maximizes the norm of the gradient loss improves the long-term optimization rate, from the NTK limit perspective. Our logical structure is as follows.

In Section 3.2 we present the paper's assumptions, after which Sections 3.3 and 3.4 introduce the mathematical framework used throughout the paper.

To justify studying the effect of parameter-space symmetries through the lens of the NTK limit, we show in Section 4 that the NTK-limit convergence rate is preserved under parameter-space symmetry transformations. This validates the use of the NTK limit in the subsequent sections.

In Sections 5.1 and 5.2 we then demonstrate that our alignment definition serves as a reliable indicator of the optimization rate. This is followed by Section 5.3, where we show that the gradient-loss norm is proportional to our alignment measure. Together with our result from Section 4, this provides an NTK-based explanation for why the procedure described in Zhao et al. (2023) improves optimization. Maximizing the gradient-loss norm increases alignment, which correlates with faster optimization. Since the NTK limit captures many of the properties of finite-width neural networks Li et al. (2019); Yang & Hu (2020); Littwin et al. (2021), this result remains relevant for realistic neural networks.

In the Appendix, we justify and elaborate on our assumptions and notation (A, B, C); extend the notion of parameter-space symmetry to general learning systems, even those that do not originally possess such symmetries (D); provide the proofs of our results (E.1, G.1); discuss nonlinear symmetries in depth (E.2); and give details of our numerical procedures (F, G.2).

### 3.2 ASSUMPTIONS AND NOTATION

We begin by summarizing the assumptions and notation used in our analysis.

Our results apply to the class of hypothesis functions defined in Equation 1, trained with deterministic gradient descent as described in Equation 9. We assume that both the hypothesis function and the loss function are twice differentiable and that the loss is Lipschitz continuous as well. We also assume that these functions admit a parameter-space symmetry, satisfying the properties outlined in Section 3.3.

When discussing linearization rate, we do so in the context of a sequence of hypothesis functions $\{F_n\}_{n=0}^{\infty}$. We assume that all functions in this sequence are trained with the same algorithm, using the same loss and training data. While a function may admit more than one symmetry, we associate each function with a single symmetry, and assume that all of these symmetries across the sequence share the same $C$ bound specified in Equation 18.

Throughout this work, we primarily interpret $n$ as the width of a neural network. However, $n$ may represent any measure of system scale. Thus, our analysis is not limited to wide neural networks, but also applies to more general sequences of function models.

To simplify notations, we often omit the explicit index $n$ and simply refer to the networks sequence as a function, as we already did in Section 1. We also omit explicit dependencies on $x \in X'$ and $\theta$ in the functions throughout the paper.

A detailed list of assumptions, together with discussions of their possible generalizations, is provided in Appendix A.

### 3.3 OUR SYMMETRY TRANSFORMATIONS

We assume symmetry transformations of the form given in Equation 2, such that they are well-defined, and twice differentiable in a neighborhood of the parameter $\theta$ where the symmetry is applied.

We mainly discuss linear symmetry transformations, such as scaling and permutation symmetries, since these constitute the vast majority of the transformations discussed in the literature Zhao et al. (2025). However, in Appendix E.2 we also address non-linear transformations.

We define the *transformation Jacobian*, such that for any two parameter indices $\alpha, \beta = 1, \ldots, N$:

$$\Gamma_{\alpha\beta}^{-1} = \frac{\partial g_\beta(\theta)}{\partial \theta_\alpha} . \tag{15}$$

We may assume the transformation is invertible without loss of generality, as we can restrict ourselves to a minimal set of effective parameters that locally describe the network. On this local manifold, $g$ becomes invertible, ensuring that $\Gamma^{-1}$ is invertible as well and that $\Gamma$ is well defined.

Since the hypothesis function remains invariant by definition, the transformation of the hypothesis function's derivatives under symmetry follows from the chain rule (see Appendix H.1):

$$\nabla \mapsto \Gamma\nabla . \tag{16}$$

To ensure finite-size learning steps, we impose a bound on the rescaling of the transformed gradient norm by some constant $0 < C$, for any $x \in X$:

$$\|\Gamma\nabla F\| \leq C \|\nabla F\| . \tag{17}$$

In fact, this condition must hold for all $x \in X$, not just the training subset $X'$. Otherwise, the related kernel could diverge outside the training data, causing the learning process to fail. Hence, any practical implementation of such symmetry transformations must at least implicitly require that:

$$\|\Gamma\| \leq C , \tag{18}$$

within the vector space spanned by the network's gradient inside a fixed-radius ball around the initialization.

We analyze the effect of this variable on our main results in Appendix C, and show that choosing $C \neq 1$ is dynamically equivalent to rescaling the learning rate as $\eta \mapsto C\eta$. Since gradient descent remains stable only when the largest eigenvalue of the normalized NTK matrix is bounded, we cannot choose a value of $C$ that effectively corresponds to a learning rate exceeding the stability threshold. In fact, the ideal learning rate is of the same order of magnitude as this threshold Cohen et al. (2021); Arora et al. (2022); Damian et al. (2022); Chemnitz & Engel (2025). This implies that, for any practical symmetry implementation, the effect of $C$ is equivalent to selecting a well-optimized learning rate, for which the NTK limit is known to hold in practice Li et al. (2019); Yang & Hu (2020); Littwin et al. (2021). For the same reason, when the learning rate is well chosen from the outset, $C$ must be close to 1.

In Appendix D, we show that any invertible matrix $\Gamma$ can be interpreted as a linear symmetry of a hypothesis function, by extending the function with an additional set of hyperparameters. This suggests that analyzing neural network symmetries through their associated transformation matrices is both natural and powerful.

### 3.4 Underlying Cause of the Neural Tangent Kernel Limit

Most works proving of the NTK limit heavily rely on the assumption that parameters are initialized independently and identically. While this assumption simplifies analysis, it offers limited intuition for why the limit holds. Moreover, it is unsuitable for our purposes, since symmetry transformations may completely violate it. Nevertheless, several works go beyond proving the NTK limit, aiming instead to uncover its underlying cause Liu et al. (2020); Chizat et al. (2019); Dyer & Gur-Ari (2019); Liu et al. (2022); Shem-Ur & Oz (2024).

A notable example is Liu et al. (2020), which argues that the decay of the Hessian's spectral norm as $n \to \infty$, is the driving mechanism behind linearization. Their main result shows that for wide neural networks, the Hessian norm converges to zero. More precisely, for any $x \in X'$ and $n \in \mathbb{N}$ (suppressing $x$ and $n$ from the notation to avoid clutter):

$$\left\| \eta \nabla^{\times 2} F \right\| = \|H\| = \sup_{\|v\|=1} v^T H v = O\left( \frac{1}{\sqrt{n}} \right) , \tag{19}$$

within any arbitrarily large but fixed radius $0 < R$ around initialization, in the NTK parametrization, (which is dynamically equivalent to the standard parametrization).

While they show that the Hessian norm governs changes in the kernel during training, they do not explicitly establish linearization in the sense of Equation 7, nor do they account for the role of the gradient scale. To address these gaps, we introduce the following lemma:

**Lemma 3.1** (Hessian Norm Governs Linearization Rate). Given the setup in Section 3.2, if there exists a parametrization such that:

$$\|H\| = O(m_n) , \tag{20}$$

where $m_n \to 0$ characterizes the Hessian decay rate, and the gradient is appropriately normalized at initialization, i.e.,

$$\left\| \sqrt{\eta_n} \nabla F_n \right\| = O\left(1\right), \quad \left\| \eta_n \nabla F_n \right\| = O\left(1\right) , \tag{21}$$

then the system linearizes as in Equation 7, where we replace $\frac{1}{\sqrt{n}}$, by the general Hessian decay/linearization rate $O\left(m_n\right)$.

We explain the meaning of the NTK parametrization, and why under standard initialization we can always treat the system as if Equation 7 holds, in Appendix A.6. The proof of Lemma 3.1 is given in Appendix E.1.

Since all these conditions are satisfied for wide neural networks with $m_n = \frac{1}{\sqrt{n}}$, it follows that, provided parameter-space symmetries do not disrupt them, such symmetries preserve the NTK limit.

## 4 Neural Tangent Kernel Limit Preservation under Parameter-Space Symmetry

To study neural teleportation within the NTK limit, we must first verify that parameter-space symmetry transformations preserve the linearization property of the NTK limit. This step is essential to ensure the applicability of the NTK framework to our analysis.

**Theorem 4.1** (NTK Limit Preservation). Given the setup of Lemma 3.1, after applying a parameter-space symmetry at an arbitrary training step $s' \in \mathbb{N}$ as described in Section 3.2, $\theta(s') \mapsto g\left(\theta(s')\right)$, the neural tangent kernel limit, as defined in Equation 7 with $m_n$ replacing $\frac{1}{\sqrt{n}}$, remains valid albeit with a different kernel. Meaning, for every $s \in \mathbb{N}$ and $x \in X$, the asymptotic bound remains:

$$\left\| F\left(\theta\left(s\right)\right)\left(x\right) - F^{lin}\left(s\right)\left(x\right) \right\| = O\left(m_n\right) . \tag{22}$$

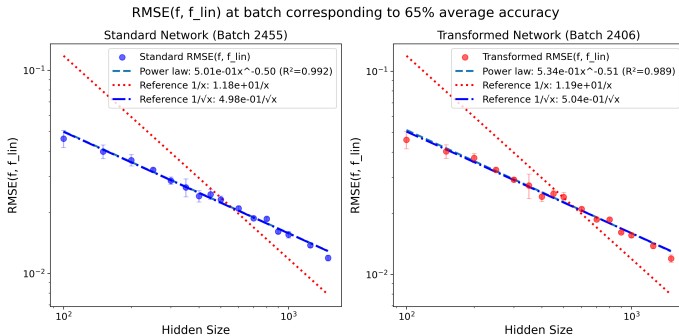

Figure 1: Root mean squared error (RMSE) between the outputs of neural networks and their linearized counterparts as a function of network width. The networks were trained on FashionM-NIST Xiao et al. (2017) for image classification. Results are averaged over three independent trials; error bars denote standard deviation. **Left:** Standard two-layer MLP, evaluated at batch 2455. **Right:** Transformed two-layer MLP, evaluated at batch 2406. Different batch indices reflect the distinct convergence speeds induced by the transformation (see App. F for details).

*Proof.* Having established the right tool for the job, the proof unfolds naturally. The key idea is to track how the Hessian and gradient norm transform under parameter-space symmetries. From Equation 16, we have:

$$H \mapsto \Gamma^T H \Gamma, \quad \nabla F \mapsto \Gamma \nabla F , \tag{23}$$

since the effect of symmetry transformations on the network's derivatives follows directly from the chain rule. This observation allows us to control the transformed gradient and Hessian norms.

Before the transformation, the gradient and Hessian norms are asymptotically bounded by Equations 21 and 20. Because the gradient norm rescaling is bounded by Equation 17, Equation 21 remains valid after the transformation. Furthermore, the Hessian spectral norm transforms as:

$$\|H\| \mapsto \left\|\Gamma^T H \Gamma\right\| = \sup_{\|v\|=1} v^T \left(\Gamma^T H \Gamma\right) v \leq C^2 \sup_{\|v\|=1} v^T H v = O\left(\tfrac{1}{\sqrt{n}}\right) . \tag{24}$$

By Lemma 3.1, this yields the desired linearization bound. $\square$

This result is unexpected, as parameter symmetry transformations can completely unravel the traditional i.i.d. assumption required for the NTK limit. Yet, since the network's outputs remain unchanged, the resulting constraints on how its derivatives transform are sufficient to ensure that the NTK limit holds.

In Appendix E.2, we extend the theorem's analysis beyond the linear case and show that the inner product of the second-derivative tensor with the gradient norm can also contribute to deviations from linearization. Since effective learning requires networks neither to deviate too far from the NTK limit nor to remain too close to it Jacot et al. (2022), we suggest that this term may serve as a design principle for constructing new nonlinear symmetries that could either mitigate or promote linearization.

We empirically validate the preservation of the NTK limit under parameter-space transformations in Figure 1, demonstrating that this regime remains applicable in practice. Further details on the experimental setup and results are provided in Appendix F.

Finally, the kernel itself transforms under symmetry as follows:

$$\Theta = \eta \nabla F(\theta_0)^T \nabla F(\theta_0) \mapsto \eta \nabla F(\theta_0)^T \Gamma^T \Gamma \nabla F(\theta_0) . \tag{25}$$

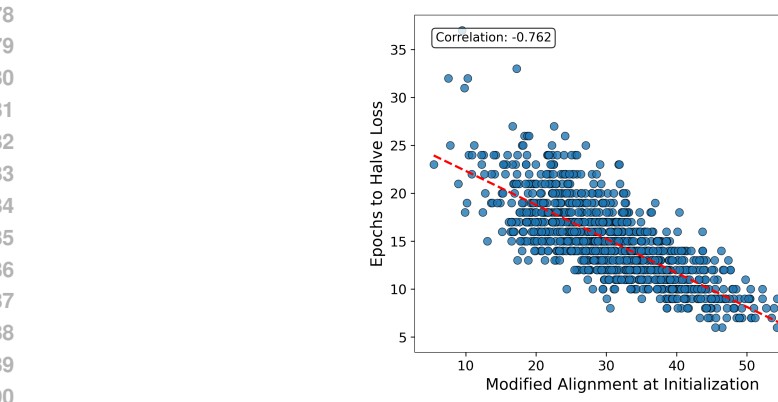

Figure 2: Relationship between modified alignment and early optimization speed. Each dot represents one of the 1,000 independently initialized MLPs. The $x$-axis shows the modified alignment measured at $t = 0$; the $y$-axis shows the epoch at which the training loss drops to one half of its initial value. The dashed line is the least-squares fit, with Pearson correlation $r = -0.76$, demonstrating that larger initial alignment values consistently lead to faster loss reduction.

## 5 OPTIMIZATION ACCELERATION

### 5.1 OPTIMIZATION RELATION TO KERNEL ALIGNMENT

As discussed in the introduction, one of the primary applications of parameter-space symmetries is to accelerate optimization. This is achieved by maximizing the norm of the empirical loss gradient as defined in Equation 10, via symmetry transformations:

$$\max \left\| \nabla \mathcal{L} \left( \theta_0 \right) \right\|^2 , \tag{26}$$

Zhao et al. (2023) demonstrated that such a transformation can be effectively applied once prior to learning, offering a computationally inexpensive way to accelerate optimization. While effective, the mechanisms by which this method enhances optimization throughout training remain poorly understood. We argue that since symmetry transformations preserve the NTK limit, this phenomenon can be analyzed through the lens of the NTK limit, by examining how such transformations affect the kernel.

A key indicator of a kernel's capacity to learn, is its alignment with the data, traditionally defined as the normalized inner product between the kernel matrix and the data vector Cortes et al. (2012); Baratin et al. (2021); Arora et al. (2019); Wang et al. (2022); Kopitkov & Indelman (2020); Khalafi et al. (2023):

$$A_o = \frac{y \left( X' \right)^T \Theta \left( X', X' \right) y \left( X' \right)}{\left\| y \left( X' \right) \right\|_2^2 \left\| \Theta \left( X', X' \right) \right\|} . \tag{27}$$

We propose that the reason gradient norm maximization accelerates learning in the NTK regime, is that it is equivalent to maximizing the kernel alignment.

### 5.2 OUR MODIFIED ALIGNMENT

To better capture the effect described above, we slightly modify the standard definition of alignment. First, instead of aligning with the data directly, we follow Kopitkov & Indelman (2020); Khalafi et al. (2023) in considering alignment with the gradient of the loss function with respect to the model output:

$$f = \mathcal{C}' \left( F, y \right) , \tag{28}$$

as $f$ represents the target signal that the NTK is actually learning, as seen in the NTK equation of motion (Equations 11,14).

Second, we remove the kernel norm from the denominator, following Khalafi et al. (2023), for two reasons: (i) up to a certain scale, larger kernels improve optimization Lewkowycz et al. (2020); Hayou et al. (2022); and (ii) in practical applications of symmetry transformations, kernel scaling must remain bounded, as discussed in Appendix A.6.

With these modifications, we redefine alignment as:

$$A = \frac{f\left(X'\right)^T \Theta\left(X', X'\right) f\left(X'\right)}{\left\|f\left(X'\right)\right\|_2^2} \; . \tag{29}$$

We establish the effectiveness of this definition in Proposition G.1, by showing that a large initial kernel alignment is a necessary condition for achieving rapid optimization.

Another theoretical justification for the relevance of our alignment definition can be naturally derived from the work of Khalafi et al. (2023). In their analysis, the kernel norm is already omitted from the denominator, for reasons similar to those discussed above. A significant part of their effort is dedicated to addressing the complications that arise from using $y$ instead of $f$. However, by directly assuming alignment with $f$, as we do, these complications are immediately avoided. The same reasoning extends naturally to any related analyses in the literature.

Additionally, we demonstrate empirically that our modified alignment reflects the optimization rate. Specifically, we train 1,000 independently initialized, fully connected ReLU MLPs with two hidden layers (1,024 neurons in each layer) and using the NTK parameterization. For each network, we compute the modified alignment at initialization and record the first epoch at which the training loss halves. As illustrated in Figure. 2, the two quantities are strongly anticorrelated ($r = -0.76$), meaning that larger initial alignment values consistently translate into faster optimization. Full experimental details are provided in Appendix G.2.

### 5.3 ALIGNMENT AND NORMED LOSS GRADIENT EQUIVALENCE

With our new definition for alignment as an indicator for optimization rate, we can now show its equivalence to the gradient of the normed loss.

**Proposition 5.1** (Alignment and Normed Loss Gradient Equivalence). Under the setup detailed in Section 3.2, the gradient of the loss function norm (Equation 26), and the kernel alignment (Equation 29), are proportional:

$$\left\|\nabla\mathcal{L}\right\|_2^2 = \frac{\left\|f\left(X'\right)\right\|^2}{\eta\left|X'\right|^2} A \propto A \; . \tag{30}$$

***Proof.*** This follows directly from Equations 8, 10, and 29, applying the chain rule within our analytical framework:

$$\left\|\nabla\mathcal{L}\right\|_2^2 = \left\|\nabla \frac{1}{|X'|} \sum_{x \in X'} \mathcal{C}\left(F\left(x\right), y\left(x\right)\right)\right\|_2^2 =$$
$$\left\|\frac{1}{|X'|} \sum_{x \in X'} \nabla F\left(x\right) f\left(x\right)\right\|_2^2 = \frac{1}{\eta|X'|^2} f\left(X'\right)^T \Theta\left(X', X'\right) f\left(X'\right) = \frac{\left\|f\left(X'\right)\right\|^2}{\eta|X'|^2} A \; . \tag{31}$$

$\square$

This concludes our explanation of why maximizing the normed loss gradient improves optimization from the NTK perspective, as it is equivalent to maximizing kernel alignment, which is correlated with the optimization rate.

## 6 CONCLUSION

In this work, we explored the effects of applying parameter-space symmetries in neural network optimization through the lens of the NTK limit. We established that such transformations preserve the NTK limit, thereby validating the use of this framework for our analysis. Leveraging this result, we showed that even a single application of parameter-space symmetry can meaningfully improve the optimization rate in the NTK regime.

Our findings suggest potential directions for the development of new optimization algorithms. In particular, they highlight more direct approaches to maximizing kernel alignment through parameter-space symmetries, or other related methods. By clarifying why parameter-space symmetrizes work, our work may help inspire new algorithms that achieve similar benefits without requiring the costly optimization often associated with parameter-space symmetries. Furthermore, our results for nonlinear symmetries may guide the design of new, useful classes of symmetries tailored for optimization.

Looking forward, we aim to extend this line of work to study how parameter-space symmetries affect generalization, through the NTK framework. By combining insights on both optimization and generalization, we hope to develop approaches that simultaneously accelerate training and improve the generalization capacity of neural networks.

A primary limitation of our work is that our analysis is restricted to the NTK limit, which remains an approximation of finite neural networks. Addressing this limitation in future work, we plan to investigate the effects of parameter-space symmetries on higher-order approximations and conduct more extensive empirical studies. Nevertheless, as the NTK framework captures many of the most relevant properties of neural networks, we believe that our findings hold meaningful value, and could facilitate practical advancements in the field.

Our work is also significant to the general understanding of the NTK limit, beyond the context of parameter-space symmetries. We provide a clear example of wide neural networks that do not satisfy what is considered a key condition of the NTK limit, namely, weak parameter correlations at initialization, yet still exhibit linearization.

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

# A  ASSUMPTIONS AND GENERALIZATIONS

In this part of the appendix, we elaborate on Section 3.2, detailing and justifying our assumptions, and discuss their generalizations.

## A.1  HYPOTHESIS FUNCTIONS

Our results apply to the function class described in Equation 1, such that they are twice differentiable everywhere. This assumption can be relaxed to cover ReLU-like networks, where functions are twice differentiable only almost everywhere, by employing a generalization of the mean value theorem, which we refer to as the Barycentric Mean Value Corollary (Corollary H.2), as discussed in Appendix E.1. This generalization allows us to include almost any practical loss function and neural network architecture.

When analyzing linearization rate, we consider a sequence of hypothesis functions $\{F_n\}_{n=1}^{\infty}$ from this class. Each $F_n$ depends on a corresponding parameter vector $\theta^n \in \mathbb{R}^{N_n}$, and each network inherits its own symmetry while being trained under the same algorithm.

The primary interpretation of $n$ will be the network width. However $n$ can represent any parameter that governs the scale of the system. For notational simplicity, we will often omit the explicit index $n$ and refer to the network as a single function rather than a sequence.

This framework is extremely general and encompasses essentially all relevant neural network architectures and beyond.

## A.2  SYMMETRY TRANSFORMATION

Our symmetries, as defined in Equation 2, are assumed to be twice differentiable and well-defined around the parameter values at which they are applied. Our main result, Theorem 4.1, is primarily applicable to linear symmetries. While this may seem restrictive, linear symmetries are the main class of transformations currently studied in the literature Zhao et al. (2025). Moreover, in Appendix E.2, we extend our discussion to the generalization of this assumption.

When analyzing how these symmetries affect the linearization rate, we further assume they satisfy the bound described in Equation 18 uniformly for every $n \in \mathbb{N}$. This condition must hold for any practical neural network transformation; otherwise, the first-order term of gradient descent would diverge, as we further discuss in Appendix C. Works such as Zhao et al. (2023) implicitly adopt this assumption by restricting the maximization search for the optimal transformation to a limited number of steps.

Additionally, we assume that the transformation Jacobian defined in Equation 15 is invertible. Otherwise, we can instead define a new effective set of parameters by projecting the original parameters onto a subspace where the transformation becomes invertible. The analysis then proceeds in the same way.

## A.3  HESSIAN SPECTRAL NORM BOUND

For Theorem 4.1, we assume that the spectral norm of the Hessian before transformation is bounded as specified in Equation 20. This follows from the arguments presented in Liu et al. (2020), as discussed in the introduction, where it was shown that the scale of the Hessian norm governs linearization, and that for wide neural networks the Hessian norm vanishes as $O\left(1/\sqrt{n}\right)$.

While Liu et al. (2020) demonstrated this behavior for a wide variety of networks, there remains a significant class of networks not explicitly covered. Nevertheless, because their proof relies on the semi-linear structure of wide networks, it is highly plausible that the linearization property extends to any network describable by the tensor program formalism Yang (2019b; 2020); Yang & Littwin (2021). Since this formalism captures practically any wide neural network, we believe this assumption to be almost certainly valid, although a full formal proof remains outstanding.

### A.4 GRADIENT DESCENT

We operate within the framework of deterministic gradient descent, as defined in Equation 9. While this seems very restrictive, all of our results naturally extend to stochastic gradient descent, and gradient descent with momentum. The analysis remains essentially the same, except that the results hold in expectation over the inputs. This transition is standard in NTK analyses.

### A.5 LOSS FUNCTION

We assume that the loss function takes the form given in Equation 10, and that it is convex, twice differentiable, and Lipschitz continuous. These are standard assumptions commonly imposed on loss functions.

### A.6 GRADIENT NORMALIZATION AND NTK PARAMETRIZATION

For Lemma 3.1, and Theorem 4.1, we assume that the gradient norm is properly normalized according to Equation 21.

The left-hand side of this equation represents the necessary condition for the kernel not to diverge. Hence, it must hold under practical applications of gradient descent with reasonable assumptions, and it is automatically satisfied for any system that admits the NTK limit. The right-hand side, by contrast, depends on the choice of parametrization.

Different parametrizations of the learning dynamics can be obtained by rescaling the parameters and then compensating by multiplying them externally by the same constant. Such transformations do not affect the training dynamics (provided the learning rate is scaled appropriately), but they allow us to view the system from different perspectives. To illustrate this idea, we consider the example of fully connected neural networks as introduced in Jacot et al. (2018).

Under the *standard parametrization*, defined in Equation 3, the initialization of parameters for all layers $l = 0, \ldots, L$ is given by:

$$\theta^{(l+1,l)} \sim \frac{1}{\sqrt{n_l}}, \quad \theta^{(l)} \sim 1, \tag{32}$$

and the learning rate scales as:

$$\eta \sim \frac{1}{n}. \tag{33}$$

In contrast, the *NTK parametrization* is defined as:

$$F^{(1)} = \frac{1}{\sqrt{n_1}}\theta^{(1,0)}x + \theta^{(1)},$$
$$\forall l = 1, \ldots, L : F^{(l+1)} = \frac{1}{\sqrt{n_l}}\theta^{(l+1,l)}\phi\left(F^{(l)}\right) + \theta^{(l+1)}, \tag{34}$$

with initialization for parameters at all layers $l = 0, \ldots, L$:

$$\theta^{(l+1,l)} \sim 1, \quad \theta^{(l)} \sim 1, \tag{35}$$

and a learning rate scaling typically given by:

$$\eta \sim 1. \tag{36}$$

Both parametrizations yield equivalent networks with respect to relevant quantities such as the kernel and the training dynamics. This reparametrization trick can always be applied to learning algorithms (though in some cases multiple learning rates may be required, which is only a technical complication) Chizat et al. (2019). It is in the NTK parametrization, where $\eta \sim 1$ and the gradient norm is $O(1)$, that the result of Liu et al. (2020), on which we rely, was established.

## B NEURAL TANGENT KERNEL LIMIT NOTATIONS

In this appendix we clarifying, and expanding upon the primary notations introduced in Section 2.2.

In the context of deterministic gradient descent, defined in Equation 9, replacing a function input with the set $X'$, signifies that we substitute the function with a vector, by combining the input from $X'$ with the output index of the original function.

Considering for example the derivative of the relative loss function from Equation 11. For any $i, j = 1, \ldots, d_Y$ and $x, a \in X'$, we can express the derivative of the loss as:

$$\mathcal{C}'\left(F\left(\theta\right)\left(X'\right), y\left(X'\right)\right)_{(i,x)} = \mathcal{C}'\left(F\left(\theta\right)\left(x\right), y\left(x\right)\right)_i . \tag{37}$$

Similarly, for the kernel with one fixed input as in Equation 11:

$$\Theta\left(x, X'\right)_{i,(j,a)} = \Theta\left(x, a\right)_{ij} . \tag{38}$$

And for the kernel with no fixed inputs, as in Equations 27 and 29:

$$\Theta\left(X', X'\right)_{(i,x),(j,a)} = \Theta\left(x, a\right)_{ij} . \tag{39}$$

Neglecting terms of order $O\left(\frac{1}{n}\right)$, we can now explicitly expand Equations 6, 8, 9, 11, detailing all indices, variables, and summations. For any $i, j = 1, \ldots, n_{L+1}$ and $x, y \in X$, we have:

$$F_i\left(\theta\right)\left(x\right) = F_i\left(\theta_0\right)\left(x\right) + \sum_{\alpha=1}^{N} \nabla_\alpha F_i\left(\theta_0\right)\left(x\right)\left(\theta - \theta_0\right)_\alpha , \tag{40}$$

$$\Theta_{ij}\left(x, y\right) = \eta \sum_{\alpha=1}^{N} \nabla_\alpha F_i\left(\theta_0\right)\left(x\right) \nabla_\alpha F_j\left(\theta_0\right)\left(y\right) , \tag{41}$$

$$\forall \alpha = 1, \ldots, N : \quad \Delta\theta_\alpha = -\eta \nabla_\alpha \mathcal{L}\left(\theta\right) , \tag{42}$$

$$\Delta F_i\left(\theta\right)\left(x\right) = -\frac{1}{|X'|} \sum_{x' \in X'} \sum_{k=1}^{n_L} \Theta_{ik}\left(x, x'\right) \mathcal{C}'\left(F_k\left(\theta\right)\left(x'\right), y_k\left(x'\right)\right) . \tag{43}$$

Here, $\theta_\alpha$ denotes the $\alpha$-th component of the parameter vector, with $\alpha = 1, \ldots, N$, where $N$ is the total number of parameters. The operator $\nabla_\alpha = \nabla_{\theta_\alpha}$ denotes differentiation with respect to the $\alpha$-th parameter. The Jacobian of the network at initialization is denoted as:

$$\nabla_\alpha F_i\left(\theta_0\right)\left(x\right) = \left.\nabla_\alpha F_i\left(\theta\right)\left(x\right)\right|_{\theta=\theta_0} . \tag{44}$$

## C  SYMMETRY BOUND EFFECT

In this section we justify the uniform bound in Equation 18 and explain how choosing $C \neq 1$ could affect the linearization rate. Our basic argument is that its influence on the dynamics is no greater than that of rescaling the learning rate by a factor of $C$. Since excessively large learning rates lead to instability, and because the NTK limit remains relevant even at the largest stable (and often optimal) learning rates, the effect of $C$ cannot invalidate our results.

The main concern is that $C$ appears to scale the Hessian norm as $C^4$. This arises from the chain-rule factor of $C^2$ in Equation 24. Since we now consider convergence not only as a function of $n$ but also of $C$, we must note that the radius of the ball on which $H$ is fixed shrinks by a factor of $C$. Thus, the effective scale of $H$ increases by another factor of $C^2$ which gives us the $C^4$ factor. This is also reflected by the fact that the gradient of $F$ is scaled by $C$ due to the symmetry. To verify this formally, one can repeat the analysis in the proof of Lemma 3.1 in Appendix E step by step, but now keeping track of the dependence on $C$. At first glance this may appear problematic, as even if we still have $O\left(m_n\right)$ convergence, the fourth-power scaling seems large enough that moderately large values of $C$ could disrupt linearization. However, this is not the case.

We can see this directly from the gradient descent update equation. Applying the linear symmetry transformation $\theta \mapsto \Gamma^{-1}\theta$, where $\|\Gamma\| = C$, Equation 9 becomes

$$\Delta\theta = -\eta\,\Gamma\nabla\mathcal{L}\left(\theta\right) = -\left(C\eta\right)\frac{\Gamma}{C}\nabla\mathcal{L}\left(\theta\right) . \tag{45}$$

Since all effects on the training dynamics arise from the gradient as the function itself is unchanged, we see that using a symmetry of norm $C$ is equivalent to using a symmetry of norm 1 together with a rescaled learning rate.

Because large but practical learning rates do not break the NTK limit, we conclude that for any realistic implementation of the NTK dynamics, varying $C$ cannot degrade the NTK regime any more than choosing an appropriately large (but still stable) learning rate. This is fully compatible with our results.

## D    GENERALIZED SYMMETRIES

### D.1    HYPOTHESIS FUNCTION EXTENSION

Here we demonstrate that any reversible matrix transformation $\Gamma$, as defined in Equation 15, can be regarded as a symmetry for any hypothesis function, as noted in Section 3.3.

To formalize this, consider a hypothesis function from the class described in Equation 1. We define its *extension* as:

$$\tilde{F}\left(\tilde{\Gamma},\theta\right) = F\left(\tilde{\Gamma}\theta\right) , \tag{46}$$

where $\tilde{\Gamma}$ is a reversible matrix.

Setting $\tilde{\Gamma} = I$ recovers the original hypothesis function. Importantly, the new hyperparameter $\tilde{\Gamma}$ is *not* updated during learning; rather, it defines a transformation that induces a symmetry.

Starting from an initial choice $\tilde{\Gamma} = I$, any subsequent reversible matrix $\Gamma$ defines a symmetry, provided we simultaneously transform the parameters as:

$$\tilde{\Gamma} = \Gamma, \quad \theta \mapsto \Gamma^{-1}\theta . \tag{47}$$

Verifying that this defines a symmetry is straightforward:

$$\tilde{F}\left(I,\theta\right) \mapsto \tilde{F}\left(\Gamma,\Gamma^{-1}\theta\right) = F\left(\Gamma\Gamma^{-1}\theta\right) = F\left(\theta\right) = \tilde{F}\left(I,\theta\right) . \tag{48}$$

Generalizing to arbitrary initializations of $\Gamma$ is similarly simple but unnecessary for our purposes. Since we only apply the transformation once during the learning process, it suffices to assume $\Gamma = I$ initially.

If we wish to restrict ourselves to vector transformations, we can replace the matrix $\Gamma$ with a vector $c$ and define the extension and corresponding symmetry using the Hadamard (element-wise) product:

$$\tilde{F}\left(c,\theta\right) = F\left(c \odot \theta\right) \quad , \quad \theta \mapsto c^{-1} \odot \theta . \tag{49}$$

### D.2    EXAMPLE - FULLY CONNECTED NEURAL NETWORKS

In the case of fully connected neural networks, as with most architectures, the parameter vector $\theta$ is structured into a sequence of matrices and vectors (as shown in Equations 3 and 4). Consequently, to preserve the structure of the Hadamard product under the transformation, the hyperparameter vector $c$ must likewise be organized into a corresponding sequence:

- Hyperparameter matrices: $c^{(1,0)} \in \mathbb{R}^{n_0 \times n_1}, \dots, c^{(L+1,L)} \in \mathbb{R}^{n_{L+1} \times n_L}$,
- Hyperparameter vectors: $c^{(0)} \in \mathbb{R}^{n_0}, \dots, c^{(L+1)} \in \mathbb{R}^{n_{L+1}}$.

To extend the network, we simply replace each parameter $\theta_\alpha$ with $c_\alpha \odot \theta_\alpha$. Thus, Equation 3 becomes:

$$\begin{aligned} F^{(1)} &= c^{(1,0)} \odot \theta^{(1,0)}x + c^{(1)} \odot \theta^{(1)}, \\ F^{(l+1)} &= c^{(l+1,l)} \odot \theta^{(l+1,l)}\phi\left(F^{(l)}\right) + c^{(l+1)} \odot \theta^{(l+1)} , \end{aligned} \tag{50}$$

where $\odot$ denotes the element-wise product between matrices or vectors.

For the linear symmetries described for fully connected networks in Section 2.1, it can be verified that all derivatives of the network after applying the real symmetry, and those of the extended network

after applying the generalized symmetry, coincide. This means, that in case where the network is analytical, the generalized network and the original network are functionally the same.

This supports the idea that generalized symmetries capture and extend the notion of symmetry for any transformation Jacobian $\Gamma$, including cases that would not naturally arise from the structure of the function itself. This perspective connects to recent architectural approaches such as graph metanetworks Lim et al. (2023) and equivariant representations of neural networks Kofinas et al. (2024).

# E  WIDE NEURAL NETWORKS LINEARIZATION

In this section we prove Lemma 3.1 and discuss the generalization of Theorem 4.1 for nonlinear symmetries.

## E.1  LEMMA 3.1 PROOF

We begin by proving the lemma.

We proceed by induction. Assume that for some step $s - 1 \in \mathbb{N}^0$ there exists a sufficiently large $0 < R$ (fixed in $n$) such that, for any $x \in X'$, the following conditions hold:

1. Lazy training:
$$\|\theta - \theta_0\| = O(1). \tag{51}$$

2. Gradient stability:
$$\|\nabla\mathcal{L}(\theta(s-1)) - \nabla\mathcal{L}(\theta_0)\| = O(m_n) = \|\nabla F(\theta(s-1)) - \nabla F(\theta_0)\|. \tag{52}$$

3. Bounded outputs:
$$O(F(\theta(s-1))) = O(1) = \mathcal{C}'(F(\theta(s-1)), y) = f(\theta(s-1)). \tag{53}$$

4. Linearization:
$$\|F(\theta(s-1)) - F_{\text{lin}}(s-1)\| = O(m_n). \tag{54}$$

By showing that these conditions also hold for the $s$-th step, and noting that the base case is trivial, we complete the proof by induction, as condition 4 corresponds to the statement of the lemma. For simplicity, we denote $\theta(s) = \theta$.

**Lazy training:**

We first show that $\theta$ remains within a fixed-radius neighborhood of initialization.

From Equation 52, and the NTK parametrization (together with Equation 21), and since the loss is Lipschitz continuous, we have:
$$\eta = O(1), \quad \mathcal{L}(\theta_0) = O(1). \tag{55}$$

Hence,
$$\eta\nabla\mathcal{L}(\theta(s-1)) = O(1) + O(m_n) = O(1). \tag{56}$$

After one gradient descent step (Equation 9):
$$\theta = \theta(s-1) - \eta\nabla\mathcal{L}(\theta(s-1)) = O(1). \tag{57}$$

Thus, there exists a neighborhood $R(s)$ (fixed in $n$), such that $\theta$ remains inside this region for almost every $n \in \mathbb{N}$.

**Gradient stability:**

We now show that gradient changes inside the $R$-neighborhood are bounded by the Hessian norm.

For any $\theta$ in this neighborhood and unit vector $v \in \mathbb{R}^N$, we define:
$$D(\delta) = v^T\sqrt{\eta}\nabla\mathcal{L}(\theta + \delta(\theta_0 - \theta)), \quad \delta \in (0,1). \tag{58}$$

By the mean value theorem, there exists $\delta' \in (0, 1)$ such that:

$$v^T \nabla \mathcal{L}(\theta) - v^T \nabla \mathcal{L}(\theta_0) = \frac{v^T}{\eta} H(\theta + \delta'(\theta_0 - \theta)). \tag{59}$$

Since $\|\theta - \theta_0\| \leq R$, the intermediate point lies within the $R$-neighborhood, so the Hessian norm is bounded by $O(m_n)$. And as $\|v\| = 1$ and $\eta = O(1)$ (Equation 21), it follows that:

$$v^T \sqrt{\eta}\big(\nabla \mathcal{L}(\theta) - \nabla \mathcal{L}(\theta_0)\big) = O(m_n). \tag{60}$$

Since this holds for any $v$, we conclude:

$$\|\nabla \mathcal{L}(\theta) - \nabla \mathcal{L}(\theta_0)\| = O(m_n). \tag{61}$$

From Equation 10, the loss gradient relates to that of the function:

$$\nabla \mathcal{L}(\theta) = \frac{1}{|X'|} \sum_{x \in X'} \nabla F(\theta)\, \mathcal{C}'(F(\theta)(x), y(x)). \tag{62}$$

Since this must hold for all $X' \subseteq X$, and the loss is Lipschitz, it follows that:

$$\|\nabla F(\theta)\| = O(m_n), \tag{63}$$

as $F = O(1)$ by Equation 54.

**Linearization.**

Similarly to before, for any unit vector $v \in \mathbb{R}^{d_y}$ and $\theta$ in the neighborhood, we define:

$$d(\delta) = v^T \big(F(\theta + \delta(\theta_0 - \theta)) + \delta \nabla F(\theta_0)^T (\theta - \theta_0)\big). \tag{64}$$

By the mean value theorem, there exists $\theta'$ in the neighborhood such that:

$$v^T \big(F(\theta) - F(\theta_0) - \eta \nabla F(\theta_0)^T (\theta_0 - \theta)\big) = v^T (\nabla F(\theta') - \nabla F(\theta_0))^T (\theta - \theta_0). \tag{65}$$

Using gradient stability, this gives:

$$\|F(\theta) - F(\theta_0) - \nabla F(\theta_0)^T (\theta_0 - \theta)\| = O(m_n). \tag{66}$$

Now, considering the network update under gradient descent (Equation 9), and applying the induction assumptions, we obtain that up to an order of $O(m_n)$:

$$\begin{aligned}
F(\theta(s)) = F(\theta(s-1) - \eta \nabla \mathcal{L}(s-1)) &\simeq F(\theta_0) - \eta \nabla F(\theta_0)^T \nabla \mathcal{L}(s-1) = \\
F(\theta_0) - \frac{\eta}{|X'|} \nabla F(\theta_0)^T \nabla F(\theta(s-1))(X')\, f(s-1)(X') &\simeq \\
F(\theta_0) - \frac{\eta}{|X'|} \Theta_0(X')\, f_{lin}(s-1)(X') = F_{lin}(s)\,.
\end{aligned} \tag{67}$$

Thus, Equation 54 holds for the $s$-th step as well.

Finally, Equation 53 follows since the NTK learning update is $O(1)$ by definition, the difference between $F$ and $F_{\text{lin}}$ is less than $O(1)$, and the loss is Lipschitz.

Completing the proof of the Lemma.

If the network $F$ is not differentiable at a finite number of points, we can instead invoke a generalization of the mean value theorem, namely the Barycentric Mean Value Corollary, proven in Appendix H.2. This theorem asserts that, instead of evaluating the derivative at a single intermediate point, the difference can be expressed as a barycentric (convex) combination of multiple intermediate points. The proof follows similarly under this extension.

E.2   THEOREM 4.1 NONLINEAR CASE

In Section 4, we proved that parameter-space symmetries do not alter the conditions of Lemma 3.1 in the case of linear symmetries. Here we extend this discussion to nonlinear symmetries.

We define the second-derivative tensor of the network as:

$$T_{\alpha\beta\gamma} = \partial_\alpha \partial_\beta g_\gamma\,, \tag{68}$$

which captures the deviation of the symmetry from linearity.

For nonlinear symmetries, the Hessian transforms as:

$$H_{\alpha\beta} = \partial_\alpha \partial_\beta \mathcal{L} \mapsto \Gamma_{\alpha\mu} \partial_\mu \Gamma_{\beta\nu} \partial_\nu \mathcal{L} = \left( \Gamma_{\alpha\mu} \Gamma_{\beta\nu} \right) \partial_\mu \partial_\nu \mathcal{L} + \Gamma_{\alpha\mu} \left( \partial_\mu \Gamma_{\beta\nu} \right) \partial_\nu \mathcal{L} \,, \tag{69}$$

where we use Einstein notation. The first term is simply $\Gamma^T H \Gamma$, which has already been analyzed. The second term requires further examination.

We connect this second term to $T$ as follows. Starting from:

$$\Gamma^{-1}_{\alpha\beta} \Gamma_{\beta\nu} = \delta_{\alpha\nu} \,, \tag{70}$$

differentiating with respect to $\theta_\mu$ gives:

$$\partial_\mu \Gamma^{-1}_{\alpha\beta} \Gamma_{\beta\nu} + \Gamma^{-1}_{\alpha\beta} \partial_\mu \Gamma_{\beta\nu} = 0 \,. \tag{71}$$

Multiplying on the left by $\Gamma_{\gamma\alpha}$ yields:

$$\partial_\mu \Gamma_{\beta\nu} = - \Gamma_{\beta\rho} \left( \partial_\mu \Gamma^{-1}_{\rho\sigma} \right) \Gamma_{\sigma\nu} \,. \tag{72}$$

Combining with Equation 15 we obtain:

$$\partial_\mu \Gamma^{-1}_{\rho\sigma} = \partial_\mu \partial_\rho g_\sigma = T_{\mu\rho\sigma} \,. \tag{73}$$

Thus:

$$\partial_\mu \Gamma_{\beta\nu} = - \Gamma_{\beta\rho} T_{\mu\rho\sigma} \Gamma_{\sigma\nu} \,. \tag{74}$$

Hence, the deviation from linearization introduced by this term is proportional to:

$$\| T \cdot \nabla \mathcal{L} \| \,, \tag{75}$$

since the norm of $\Gamma$ is bounded by a constant $C$, independent of $n$.

Therefore, for the neural tangent kernel limit to be preserved under nonlinear symmetries, this additional norm must also remain small. This provides a practical guideline for designing nonlinear symmetries: effective learning requires avoiding excessive deviation from NTK dynamics, which in turn necessitates controlling the growth of this nonlinear term. Conversely, if the objective is to move away from the NTK regime, since being too close to it can also hinder learning, this term may be deliberately exploited to induce such deviation.

## F  EXPERIMENTAL VERIFICATION OF THE NTK LIMIT PRESERVATION

In this appendix, we demonstrate that the studied transformations preserve the NTK limit. We implemented our experiments in Python using JAX, leveraging the Neural Tangents library Novak et al. (2019) with a customized Stax library. All experiments use fully-connected feedforward networks (MLPs) with the *NTK parametrization*, trained on the FashionMNIST dataset Xiao et al. (2017). All computations were performed on a single NVIDIA GTX 4090 GPU.

To verify the NTK limit for standard and transformed networks, we tracked the root mean squared error (RMSE) between the outputs of a neural network and its linearized counterpart. This metric is known to decrease as the network width increases, following a power law of $\frac{1}{\sqrt{n}}$, where $n$ is the width of the hidden layer.

We used networks with two hidden layers of equal width $n$, ranging between $n = 100$ and $n = 1500$ with varying steps. The standard network was initialized using the NTK parametrization, and for simplicity, biases were omitted. The transformed network was obtained by using the generalized symmetry, as described in Section D, with scaling hyperparameters $c^{(l+1,l)}$. The generalized symmetry transformation consists of scaling each parameter at initialization, and compensating for that through the architectural change (see below). Specifically, each initialized weight was divided by the corresponding hyperparameter $c^{(l+1,l)}$, which remained fixed throughout training. These hyperparameters were obtained through an optimization process with the objective of maximizing the empirical loss gradient norm $\|\nabla L(f(x), y)\|$. Starting with uniform values (all ones), the method employs Adam optimizer with learning rate $\eta = 0.01$ to iteratively update these hyperparameters. To maintain

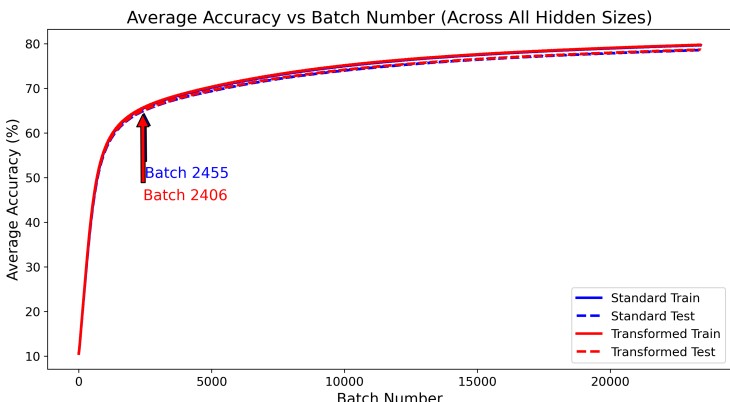

Figure 3: Test accuracy of networks, averaged across all hidden sizes, over training iterations (batches). The specific batch at which each network reached 65% test accuracy is highlighted. The faster convergence speed for the transformed network is caused by the hyperparameters $c^{(l+1,l)}$ being optimized and larger than one on average. Results are averaged over three independent trials.

stability, the optimization includes regularization that penalizes large deviations from initial values via a term $\lambda(\|c - 1\|^2)$, where $\lambda = 0.1$ was used. Additional constraints include gradient clipping to prevent large updates and explicit bounds ($0 \leq c_i \leq c_{\max} = 10$) on hyperparameter values. This approach results in small deviations from initial values of hyperparameters that are under 5%.

The modification in the transformed network's architecture consisted of applying a Hadamard product between the hyperparameters and corresponding weights in the forward pass, which realizes the generalized symmetry. Finally, both networks (standard and transformed) were linearized by taking the first-order Taylor approximation and trained using stochastic gradient descent (SGD) with momentum 0.9 for 200 epochs, with a learning rate of 1.

The results are presented in Figure 1. Each data point represents the average over three independent trials, with error bars indicating the standard deviation across these trials. To account for differences in convergence speed, snapshots were taken at different batch numbers for each plot. Specifically, when the hyperparameters are larger than one, the optimization process accelerates, requiring an earlier evaluation point for a fair comparison. Pairs of networks (standard and transformed) in each trial share the same initialization up to a symmetry transformation, which induces a correlation between their outputs and partially aligns the fluctuations around the fitted scaling law.

To ensure comparability between both plots (snapshots), we selected a batch where the average accuracy across all hidden sizes reached 65% (see Figure 3). This threshold was chosen heuristically, representing a point in training where accuracy and loss fluctuations become reasonably small.

## G   MODIFIED ALIGNMENT AND TRAINING DYNAMICS

In this section we show, both numerically and analytically, that our new definition of neural tangent alignment in Equation 29 is a reliable indicator of the linearization rate.

### G.1   THEORETICAL JUSTIFICATION

We begin with the theoretical perspective, showing that if the initial alignment is small, then the training rate must also be small.

We work with the setup described in Section 2.2, in the strictly linear limit, where the training dynamics are:

$$f(s + 1) = (I - \Theta(X', X')) f(s) , \tag{76}$$

and, assuming mean squared error (MSE) loss for simplicity (we discuss generalization below), so the loss is:

$$\mathcal{L}(s) = \frac{1}{2|X'|}\,\|f(s)\|_2^2\;. \tag{77}$$

We make the following argument:

**Proposition G.1** (Small Kernel Alignment Leads to Learning Rate). Given the setup above, with the kernel alignment defined as in Equation 29, then for every $s \in \mathbb{N}$,

$$\frac{\mathcal{L}(s)}{\mathcal{L}(0)} \geq (1-A)^{2s} \geq 1 - 2sA\;, \tag{78}$$

and the exponential learning rate is bounded by:

$$2\ln\left(\frac{1}{1-A}\right) \simeq 2A + O\left(A^2\right)\;. \tag{79}$$

**Proof.** Since $\Theta(X', X')$ is symmetric positive semi-definite (PSD), diagonalize it as:

$$\Theta = U\,\Lambda\,U^T\;,\qquad \Lambda = \mathrm{diag}(\lambda_1,\ldots,\lambda_m)\;,\;\;\lambda_i \geq 0\;. \tag{80}$$

Writing $f(0) = Uv$ with $v \in \mathbb{R}^m$, the linear dynamics in equation 76 imply

$$f(s) = (I - \Theta)^s f(0) = U\,(I - \Lambda)^s v\;. \tag{81}$$

Therefore,

$$\frac{\|f(s)\|_2^2}{\|f(0)\|_2^2} = \sum_{i=1}^m w_i\,(1-\lambda_i)^{2s}\qquad \text{with}\qquad w_i := \frac{v_i^2}{\sum_{j=1}^m v_j^2}\;,\;\;w_i \geq 0\;,\;\;\sum_{i=1}^m w_i = 1\;. \tag{82}$$

For fixed $s \geq 1$, the function $g(\lambda) := (1-\lambda)^{2s}$ satisfies

$$g''(\lambda) = (2s)(2s-1)\,(1-\lambda)^{2s-2} \geq 0\;, \tag{83}$$

hence $g$ is convex on $[0, \infty)$. By Jensen's inequality applied to Equation 82,

$$\sum_{i=1}^m w_i\,g(\lambda_i) \geq g\left(\sum_{i=1}^m w_i\lambda_i\right) = \left(1 - \sum_{i=1}^m w_i\lambda_i\right)^{2s}\;. \tag{84}$$

Observe that

$$\sum_{i=1}^m w_i\lambda_i = \frac{f(0)^T\Theta f(0)}{\|f(0)\|_2^2} = A\;. \tag{85}$$

Combining Equations 82, 84, and 85 gives

$$\frac{\|f(s)\|_2^2}{\|f(0)\|_2^2} \geq \left(1-A\right)^{2s}\;. \tag{86}$$

Multiplying both sides by $1/(2|X'|)$ yields from Equation 78:

$$\frac{\mathcal{L}(s)}{\mathcal{L}(0)} \geq (1-A)^{2s} \geq 1 - 2sA\;, \tag{87}$$

Where the second The inequality then follows from Bernoulli's inequality.

Finally:

$$\frac{\mathcal{L}(s)}{\mathcal{L}(0)} \geq e^{\ln\left((1-A)^{2s}\right)} = e^{-\left(2\ln\left(\frac{1}{1-A}\right)\right)s}\;, \tag{88}$$

showing that the exponential rate is bounded by $2\ln\left(\frac{1}{1-A}\right)$. $\qquad\square$

This result extends beyond mean squared error without changing the alignment definition. For a general differentiable convex loss $\mathcal{C}$, the vector $f(s)$ is, by definition, the derivative of the loss with respect to the network outputs on $X'$ (see Equation 12). In the strictly linear NTK regime (Equation 11), a discrete update in function space reads:

$$F(\theta(s+1))(X') - F(\theta(s))(X') = -\Theta(X', X')f(s) .$$

By convexity of $\mathcal{C}$, the standard first-order upper bound gives:

$$\mathcal{L}(s+1) - \mathcal{L}(s) \leq \langle f(s), F(\theta(s+1))(X') - F(\theta(s))(X') \rangle = -f(s)^T \Theta(X', X')f(s) . \quad (89)$$

Thus, the same kernel alignment quadratic form $f(s)^T \Theta f(s)$ governs the one-step improvement for any convex loss. Evaluated at initialization, this yields the initial decrease bound in terms of $A = \frac{f(0)^T \Theta f(0)}{\|f(0)\|_2^2}$ exactly as in Proposition G.1. Moreover, the Jensen/convexity argument used in the proposition applies unchanged to the NTK-driven evolution of $f(s)$, implying that the error-signal derivatives cannot contract faster than $(1 - A)^s$ along the dynamics. Inserting this into Equation 89 shows that a small alignment $A$ entails a small training rate for general convex losses as well. Importantly, no special quadratic structure of the loss is required, only convexity and linear NTK dynamics.

## G.2 EMPIRICAL DEMONSTRATION

In this appendix we show that our modified alignment (Equation 29), when measured at initialization, indicates the network's subsequent optimization speed numerically. The experiment was implemented in Python using JAX and the neural tangents library, and executed on a single NVIDIA RTX 4090 GPU. It uses fully–connected feed-forward networks (MLPs) with two hidden layers of width $n = 1024$, ReLU activation functions, and the NTK parametrization. We omitted biases for simplicity. A subset of FASHION-MNIST is used, consisting of $N_{\text{train}} = 5000$ training and $N_{\text{test}} = 1000$ test images.

The initial alignment is obtained on the first 1,000 training samples by computing their empirical NTK and combining it with the corresponding untrained logits (based on the Equation 29). This procedure is repeated for 1,000 independent random initializations.

Each network is then trained for 100 epochs with stochastic gradient descent (learning rate 1, momentum 0.9, batch size 128), recording train and test metrics after every epoch.

To reflect the optimization speed at the early phases of training, we record the first epoch at which the training loss falls below one half of its initial value. Figure 2 plots the loss-halving epoch against the modified alignment measured at initialization.

The Pearson correlation is $r \approx -0.76$, supporting the claim that the modified alignment is a good metric for the optimization rate.

# H ADDITIONAL DERIVATIONS AND PROOFS

Here we present some of the mathematical calculations and details used throughout the paper.

## H.1 SECTION 3.3

First, we prove that the transformation Jacobian defined in Equation 15 indeed governs how the derivatives of the network transform under the symmetry, as illustrated in Equation 16.

This is shown by observing that for any two parameter indices $\alpha, \beta = 1, \ldots, N$, according to the chain rule, the gradient with respect to the transformed parameters $\nabla'$ where $\theta' = g(\theta)$ is related to the original parameters $\theta$, after the symmetry transformation as follows:

$$(\nabla' F(\theta'))_\alpha = \frac{\partial F(\theta')}{\partial \theta'_{\alpha'}} = \sum_{\beta=1}^{N} \frac{\partial \theta_\beta}{\partial \theta'_\alpha} \frac{\partial F(\theta')}{\partial \theta_\beta} = \sum_{\beta=1}^{N} \frac{\partial \theta_\beta}{\partial g_\alpha(\theta)} \frac{\partial F(g(\theta))}{\partial \theta_\beta} . \quad (90)$$

This calculation is valid because the symmetry is assumed to be well-defined and differentiable in a neighborhood of $\theta$. We can then use the fact that in this neighborhood, by definition:

$$F(g(\theta)) = F(\theta) , \quad (91)$$

which yields the desired result.

The same derivation applies to higher-order derivatives as well.

## H.2 SECTION E.1

**Corollary H.1** (Barycentric Mean Value).

Let $a < b$ and let $f : [a, b] \to \mathbb{R}$ be continuous on $[a, b]$ and differentiable on $(a, b)$ except at finitely many points. Then there exist

$$c_1, \ldots, c_N \in (a, b) \qquad \text{and} \qquad \lambda_1, \ldots, \lambda_N > 0, \ \sum_{j=1}^{N} \lambda_j = 1 \,, \tag{92}$$

such that

$$\frac{f(b) - f(a)}{b - a} = \sum_{j=1}^{N} \lambda_j \, f'(c_j) \,. \tag{93}$$

*Proof.*

Denote by $\{d_1, \ldots, d_k\} \subset (a, b)$ the (finite) set of points where $f$ fails to be differentiable and set

$$d_0 := a, \qquad d_{k+1} := b \,. \tag{94}$$

Because $f$ is continuous on $[a, b]$ and differentiable on each open sub-interval

$$I_j = (d_{j-1}, d_j) \quad (j = 1, \ldots, k+1) \,, \tag{95}$$

the classical Mean Value Theorem applies to $f$ on every $[d_{j-1}, d_j]$:

$$\exists\, c_j \in (d_{j-1}, d_j) \quad \text{s.t.} \quad f'(c_j) = \frac{f(d_j) - f(d_{j-1})}{d_j - d_{j-1}} \quad (j = 1, \ldots, k+1) \,. \tag{96}$$

Define positive weights:

$$\lambda_j := \frac{d_j - d_{j-1}}{b - a}, \qquad j = 1, \ldots, k+1 \,. \tag{97}$$

They satisfy $\sum_{j=1}^{k+1} \lambda_j = 1$ and $\lambda_j > 0$ because $d_j > d_{j-1}$.

Now compute:

$$\sum_{j=1}^{k+1} \lambda_j \, f'(c_j) = \frac{1}{b-a} \sum_{j=1}^{k+1} (d_j - d_{j-1}) \frac{f(d_j) - f(d_{j-1})}{d_j - d_{j-1}} = \frac{1}{b-a} \sum_{j=1}^{k+1} \big[ f(d_j) - f(d_{j-1}) \big] \,. \tag{98}$$

The sum telescopes:

$$\sum_{j=1}^{k+1} \big[ f(d_j) - f(d_{j-1}) \big] = f(d_{k+1}) - f(d_0) = f(b) - f(a) \,. \tag{99}$$

Hence:

$$\sum_{j=1}^{k+1} \lambda_j \, f'(c_j) = \frac{f(b) - f(a)}{b - a} \,. \tag{100}$$

Setting $N := k + 1$ completes the proof.

$\square$

