# OpenReview forum: "Neural Tangent Kernel Perspective on Parameter-Space Symmetries"
_ICLR.cc/2026/Conference — Submitted to ICLR 2026_

### Official Review · Reviewer_h7c3 · 2025-10-21

**Soundness:** 3
**Presentation:** 3
**Contribution:** 3
**Rating:** 6
**Confidence:** 4

**Summary:**

The paper studies the effects of *parameter-space symmetries* (transformations of parameters that leave the network’s output function unchanged) on neural network training. The two main contributions are: (1) the authors show that if the NTK regime (linearized training) holds for a network with initial parameters $\theta$, it also holds for a network with parameters $g(\theta)$, where $g$ is a symmetry transformation with Jacobian $\Gamma^{-1}_{ij} := \partial g_i(\theta)/\partial \theta_j$ satisfying $\||\Gamma\||\le C$; (2) they demonstrate that maximizing the norm of the initial loss gradient $\||\nabla\mathcal{L}(\theta_0\||$ is related to maximizing the alignment between the NTK and the effective residuals. The second contribution is connected to parameter symmetries through the observation that prior work on such symmetries used them to maximize $\||\nabla\mathcal{L}(\theta_0\||$ in order to speed up training.

**Strengths:**

**(S1) Framing:** The paper aims to connect several rather disjoint concepts: neural network symmetries, the NTK regime, and kernel alignment. To my mind, this is an interesting idea and I generally enjoyed reading the paper.

**(S2) NTK regime with symmetries:** The observation that convergence in the NTK regime is preserved under symmetry transformations is, to the best of my knowledge, novel and interesting. The proof based on the Hessian norm bound is brief and elegant, showing that the authors indeed identified an appropriate and effective analytical framework.

**(S3) Clarity and precision:** The results and derivations are clearly and formally stated.

**(S4) Presentation:** The writing quality is good. The main text communicates the core ideas cleanly, and refers details and heavier notation to the appendix.

**Weaknesses:**

**(W1) Missing link between the NTK regime and optimization parts:** The two main contributions (the NTK-regime invariance result and the discussion of NTK alignment and optimization) seem rather disconnected. The authors show that symmetries preserve the NTK regime and that maximizing the loss gradient norm can improve optimization through better NTK alignment. However, the paper does not clearly explain why or how the considered symmetries would actually promote such alignment, or why symmetries are a particularly effective way to achieve this compared to other methods. The connection between these parts appears only briefly, justified mainly by previous empirical work using symmetries to maximize the loss gradient. Therefore, statements such as *“we showed that even a single application of a parameter-space symmetry can meaningfully improve the optimization rate in the NTK regime”* (Conclusion) seem overstated.

**(W2) Limited experiments:** The experiments are limited to two-layer MLPs on a subset of FashionMNIST and appear to cover only one form of rescaling symmetry. It would be much more interesting to investigate additional types of symmetries with different corresponding $C$ constant, and analyze how these affect convergence to the NTK regime. This would clarify whether the theoretical analysis indeed captures the underlying mechanisms.

**(W3)** There are also a few minor issues related to clarity: (1) the NTK alignment discussion is rather vague, the same argument might be stated more clearly via spectral bias; (2) the motivation for the specific symmetry model could be explained further. In particular, it would be useful to discuss more realistic examples of parameter symmetries and explain why or when they satisfy the paper’s assumptions (e.g., invertible $\Gamma$).

**Questions:**

- Is it discussed anywhere in the paper how parameter-space symmetries actually influence the considered alignment notion?
- Could you elaborate on how the results depend on specific types of symmetries and network architectures? For instance, how does the constant $C$ vary across transformations and models?
- How do you see the main practical application or insight of your results? The paper mentions potential use in designing more efficient optimization strategies—could you e.g. elaborate on this?

---

> ### Author Response · Authors · 2025-11-27
>
> Thank you for your review. We are pleased that you found our paper novel, well written, interesting, and elegant.
>
> **Regarding your first concern:**
>
> As we understand it has two components: (i) the connection between Sections 4 and 5 is not clear enough, and (ii) the role of parameter symmetries in promoting kernel alignment is not explained thoroughly.
>
> **We will begin by adresing the second point:**
>
> > ...the paper does not clearly explain why or how the considered symmetries would actually promote such alignment, or why symmetries are a particularly effective way to achieve this compared to other methods....
>
> Our argument in Section 5 is that the symmetries promote alignment because their parameters are chosen to maximize the gradient-loss norm. Since the gradient-loss norm is proportional to kernel alignment, maximizing one is equivalent to maximizing the other. Our goal is not to propose a new optimization technique, or to claim that parameter symmetries are the best way to increase alignment. Instead, we aim to explain why the method proposed in [1], applying a symmetry once with its parameters chosen to maximize the gradient-loss norm, can speed up training.
>
> To do this, we first show in Sections 5.1 and 5.2 that our alignment measure predicts optimization rate near the NTK limit. Then, in Section 5.3, we show that maximizing the gradient-loss norm is equivalent to maximizing alignment. The specific method used to optimize the symmetry parameters is not central to the argument (although, as we note in the introduction, in [1] they are obtained by running gradient descent on the symmetry variables). What matters is that maximizing the gradient-loss norm improves optimization, because this is the same as maximizing kernel alignment, and our alignment measure is a reliable predictor of optimization rate.
>
> **Addressing your first point:**
>
> > ...The two main contributions ... seem rather disconnected...
>
> The connection between the two main contributions, i.e., Section 4 and Section 5, is that Section 5 relies on the NTK framework, and Section 4 shows that this framework remains valid even after applying parameter-space symmetries. Without Section 4, the analysis in Section 5 could have no meaning, as the NTK limit might no longer hold. We explain this motivation at the beginning of Section 4, but we agree that this link should be made more explicit.
>
> To summarize the paper’s logical structure:
>
> 1. Section 4 shows that the NTK framework still applies after applying parameter space symmetries.
>
> 2. Sections 5.1 and 5.2 show that our new alignment definition predicts optimization rate.
>
> 3. Section 5.3 ties these two part together, by proving that maximizing the gradient-loss norm is equivalent to maximizing alignment.
>
> Together, these results give an NTK-based explanation for why a single symmetry-based gradient loss norm maximization step improves optimization.
>
> Thank you for stressing the importance of presenting this structure clearly. We will add a short overview of this logical flow in Section 3 to help readers follow the paper’s argument.
>
> ---
>
> [1] Zhao, Bo, et al. "Improving Convergence and Generalization Using Parameter Symmetries." The Twelfth International Conference on Learning Representations.

---

> > ### Author Response · Authors · 2025-11-27
> >
> > **About the limited experiments:**
> >
> > > ...The experiments are limited to two-layer MLPs on a subset ... cover only one form of rescaling symmetry...
> >
> > Our experiments focus on the most fundamental neural network architecture, the fully connected neural network. The symmetry we use, described in Appendix C, is the most general form of linear symmetry, and linear symmetries constitute most of the relevant symmetries considered in practice. Including rescaling and permutations symmetries. In that sense, our empirical results actually address a more general case than almost any specific symmetry typically studied. While more restrictive symmetries could, in principle, behave differently, we currently see no reason to expect this. For a theoretical paper, we believe that the experiments we provide are sufficient as demonstrations of our results.
> >
> > **About the minor issues related to clarity**:
> >
> > > ...the NTK alignment discussion is rather vague...
> >
> > We aimed to provide clear intuition and motivation for our alignment definition and demonstrated its effectiveness both theoretically and numerically in Appendix F.
> >
> > > ...it would be useful to discuss more realistic examples of parameter symmetries... when they satisfy the paper’s assumptions (e.g., invertible $\Gamma$).
> >
> > Our formalism naturally accommodates any linear symmetry, which together constitute the main class of transformations studied in the literature, as discussed in Section 3.2 and Appendix A.2.
> >
> > Regarding the invertibility assumption of $\Gamma$, you are absolutely right that this deserves a clearer explanation. The key observation is that non-invertible symmetries occur only when certain directions in parameter space leave the network unchanged. In these cases, we can restrict our attention to the minimal set of parameters that fully characterizes the network locally. This reduction yields an effective parameterization that captures both the network and its derivatives, and therefore its local dynamics. With respect to these effective parameters, the Jacobian becomes invertible. Hence, we can always treat the network as if the invertibility condition holds.
> >
> > Thank you for pointing out the need to clarify this issue. We will expand this discussion and include a formal argument in the next revision.
> >
> > **Regarding your questions:**
> >
> > > Is it discussed anywhere in the paper how parameter-space symmetries actually influence the considered alignment notion?
> >
> > Yes. Parameter-space symmetries affect the kernel through Equation 24, and this change propagates to the alignment expression in Equation 28. Since we maximize the gradient-loss norm, and Equation 30 shows the equivalence between maximizing this quantity and maximizing alignment, the symmetries influence alignment indirectly through their effect on the kernel.
> >
> > > Could you elaborate on how the results depend on specific types of symmetries and network architectures? For instance how does the constant $C$ vary across transformations and models?
> >
> > Our results are agnostic to the specific network architecture, as long as the network operates in the NTK regime, and they apply to any symmetry that satisfies the paper’s assumptions.
> >
> > For a finely tuned learning rate, meaning a learning rate that ensures the largest eigenvalue of the normalized kernel is close to 1, the constant $C$ must also be around, or smaller than, 1. Otherwise, the kernel becomes too large and training fails. When the learning rate is not finely tuned, one can choose larger values of $C$ so that the combined effect of $C$ and the learning rate is equivalent to using a well-tuned learning rate.
> >
> > Thank you for emphasizing the importance of clarifying this point. We will include a more detailed discussion of this dependence in the next version of the text.
> >
> > > How do you see the main practical application or insight of your results? The paper mentions potential use in designing more efficient optimization strategies—could you e.g. elaborate on this?
> >
> > While the main implications of the paper are theoretical, and concern the general understanding of symmetries and the NTK limit in neural networks, we think that our results would lead to direct practical applications. In particular, when we will extend these tools to understand how parameter-space symmetries influence generalization, as we do here for optimization, this could open the door to alternative optimization strategies. Such strategies would aim not only to improve optimization rate by maximizing the gradient-loss norm, but also to improve generalization by explicitly exploiting the structure of the symmetry transformations.
> >
> > Thank you again for your review. We are glad that you found the paper to be of high quality. We hope that our responses and planned revisions address your main concerns, and that you will consider raising your score.

---

### Official Review · Reviewer_5haL · 2025-10-23

**Soundness:** 2
**Presentation:** 4
**Contribution:** 2
**Rating:** 2
**Confidence:** 4

**Summary:**

Disclaimer: I have reviewed a previous version of this paper, which was submitted at NeurIPS. I carefully read this new version. The review below contains elements from my previous review and takes into account the discussion that took place among the reviewers (who decided to reject the paper).

This paper studies parameter space symmetries for neural network training through the lens of the neural tangent kernel framework. First, the paper shows that the NTK regime holds with the same asymptotic rate (up to a constant factor) when applying symmetry transformations. This effectively changes the NTK kernel. Interestingly, the paper shows that symmetry transformations that maximize the norm of the gradient at initialization yields a kernel with a better alignment property with the data, hence accelerating optimization. It is related to a technique called neural teleportation: The goal is then to explain such an empirical phenomenon from the NTK point of view.

**Strengths:**

- The paper is very well written and easy to follow throughout, including the proofs and appendix.
- It addresses a missing link between theory and an empirical phenomenon.
- The use of a modified alignment criterion is now well justified.

**Weaknesses:**

- The central motivation is to explain the benefits of a technique known as neural teleportation, which involves applying symmetry transformations during or before training in the parameter space. While the phenomenon is interesting, I remain somewhat skeptical about its significance. Two years after its introduction, the technique still appears to be relatively marginal and has not seen widespread adoption in practice. Nevertheless, this is a minor comment and I admit that investigating this phenomenon is fully justified (I remember the answer from the authors regarding this point).

- There was however a major issue raised during the previous round of reviews, which convinced me that the paper was not ready for publication, even though I was rather positive in the first place. The O(1/sqrt{n}) has a C^4 dependence, which was explicit in the previous version of the paper, but this issue is now hidden under the carpet, C being hidden in the big O notation. Without a precise understanding of the magnitude of C, it is hard to understand whether or not the 1/sqrt{n} is meaningful. I remember an argument that without bounded C, learning is impossible. This is probably true but we are missing a precise mathematical statement here.

More precisely, assuming it is reasonable to restrict the analysis to a class of bounded transformations with constant C, achieving the rate 1/sqrt{n} may be a positive result, but the scaling with C^4 is actually rather bad (negative result). I also remember an argument stating that  C should be close to 1. This is a critical point. We are perhaps missing either a precise statement or an empirical study with real-world data/transformations (where the transformations really help) and the corresponding value of C.

**Questions:**

A discussion about the nature of C with precise mathematical statements will be crucial to change my assessment.

---

> ### Author Response · Authors · 2025-11-27
>
> Thank you for your review. We are happy that you found our paper well written and of an important contribution. We hope you have noticed that we addressed the main remaining issues from the NeurIPS submission (where, as you may remember, we already came close to acceptance even before making these changes). These revisions include providing an analytical proof that our alignment measure is an effective indicator of the optimization rate, together with an empirical demonstration in Figure F.1, as well as a detailed discussion of nonlinear transformations in Appendix D.2.
>
> Another change we have made is, as you noted, to remove the discussion of how the NTK linearization rate scales with $C$:
>
> > ...More precisely, assuming it is reasonable to restrict the analysis to a class of bounded transformations with constant $C$, achieving the rate $1/sqrt{n}$ may be a positive result, but the scaling with $C^4$ is actually rather bad (negative result)...
>
> The reason we omitted this discussion is that we found it introduced confusion and was ultimately beside the point, as increasing $C$ is effectively equivalent to increasing the learning rate. Thus, $C$ can hinder linearization only to the same extent that a large learning rate can.
>
> This can be seen as follows. Taking $C>1$ allows the gradient norm to increase by a factor of $C$, which, by the chain rule, is equivalent to requiring that none of the network’s parameters shrink by more than a factor of $C$ (up to a symmetry transformation that rotates norms across the parameter set). However, again by the chain rule, this same factor of $C$ can be absorbed into the learning rate: the dynamical update equation (Equation 11) and its higher-order expansion change in the same way if all parameters are scaled by $\frac{1}{C}$ or if the learning rate is scaled by $C$. Hence, for any $1<C$, the resulting dynamics coincide with those obtained by choosing $C=1$ and replacing $\eta$ with $C\eta$, whenever the system reaches the upper bound imposed by $C$.
>
> In practice, good optimization performance typically requires choosing a learning rate such that the largest eigenvalue of the kernel after scaling by the learning rate, remains bounded. When the learning rate becomes even modestly larger, the dynamics degrade rapidly [2-6]. Therefore, for any effective use of such symmetries in systems with highly tuned learning rates, we must choose $C$ close to 1. Otherwise, the effective learning rate becomes too large and the largest kernel eigenvalue becomes unstable.
>
> For the same reasons, in settings where the learning rate is not well tuned, choosing $C$ far from 1 is simply equivalent to using a better-adjusted learning rate. Since the NTK limit is relevant for well tuned learning rates, $C$ does not pose any additional issue beyond the usual considerations for selecting an appropriate learning rate, which is why we originally chose not to emphasize it.
>
> We appreciate your suggestion to clarify this point. For completeness, we will restore in the appendix the derivation showing how the linearization rate scales with $C$, refer to it from the main text, and explicitly explain the equivalence between choosing $C\approx 1$ and using a well-optimized learning rate.
>
> ---
>
> [2] Jacot et al., “Neural Tangent Kernel: Convergence and Generalization in Neural Networks,” NeurIPS 2018.
>
> [3] Allerbo, “Solving Kernel Ridge Regression with Gradient Descent for a Non-Constant Kernel,” 2023.
>
> [4] Cohen et al., “Gradient Descent on Neural Networks Typically Occurs at the Edge of Stability,” ICLR 2021.
>
> [5] Arora et al., “Understanding Gradient Descent on the Edge of Stability in Deep Learning,” ICML 2022.
>
> [6] Lewkowycz et al., “The Large Learning Rate Phase of Deep Learning: The Catapult Mechanism,” 2020.

---

> ### Author Response · Authors · 2025-11-27
>
> **Regarding the contributions:**
>
> > While the phenomenon is interesting, I remain somewhat skeptical about its significance...
>
> As with many emerging methods, it is natural that Neural Teleportation initially struggles to compete with well-optimized existing approaches. However, this does not preclude the possibility that, with further research, Neural Teleportation may offer significant contributions. More broadly, symmetry transformations constitute a well-established area of research, with many works dedicated to their theoretical and empirical properties (see, for example, the comprehensive review [7]). Understanding how such transformations affect learning dynamics is fundamentally important.
>
> Furthermore, a deeper theoretical understanding of Neural Teleportation may provide insights into several related topics, including matrix preconditioning, generalization, and flatness. We also believe that our work makes additional contributions: We introduce and demonstrate the efficacy of a new alignment definition, and we offer insights relevant to the broader understanding of the NTK limit beyond parameter-space symmetries. In particular, we provide a clear example of wide neural networks that violate what is typically considered a key condition for the NTK limit, namely, weak parameter correlations at initialization, yet still exhibit linearization.
>
> We thank you for your review and hope that we have addressed your concerns regarding the soundness and impact of the paper. We hope that with these changes you will consider raising your score, in light of your optimistic assessment in the previous round.
>
> ---
>
> [7] Zhao, Bo, Robin Walters, and Rose Yu. "Symmetry in Neural Network Parameter Spaces." arXiv preprint arXiv:2506.13018 (2025).‏

---

### Official Review · Reviewer_UXBr · 2025-10-30

**Soundness:** 3
**Presentation:** 3
**Contribution:** 3
**Rating:** 4
**Confidence:** 3

**Summary:**

This paper examines the reparameterization symmetries under transformations of the parameters $\theta \to \theta' = g(\theta)$ in the kernel (linearization) regime of neural network training. In this regime, the Jacobian of the transformation $\Gamma^{-1} = \frac{\partial g(\theta)}{\partial \theta}$ is also static over the course of optimization. The authors argue (by means of examining the transformed gradients and Hessians) that for any $\Gamma $ with bounded norm, the reparameterization does not prevent linearization, significantly relaxing assumptions commonly employed to induce a kernel limit. They use their theory to argue for a relationship between the kernel alignment and speed of training.

**Strengths:**

This paper examines an interesting question of how reparameterization $\theta \to \theta'$ impacts training dynamics of neural networks in the linearized regime. It also provides a nice description of training with the transformed NTK $\frac{\partial f}{\partial \theta} \Gamma^\top \Gamma \frac{\partial f}{\partial \theta}$ and characterizing the transformed Hessian.

**Weaknesses:**

**Few Example Applications of the Main Result** The authors allude to methods that reparameterize a model to speed up convergence. Could they design better initialization or reparameterization strategies? The current empirical figure simply reports a correlation between alignment and training speed.

**Questions:**

1. The authors claim that "maximizing the norm of the empirical loss gradient as defined in Equation 10, via symmetry transformations" provides a method to accelerate training. How is this done in practice? Are there analytical methods for this or is it done numerically.
2. Can you use this theory to account for the inducement of lazy training through output rescaling as in https://arxiv.org/abs/1812.07956? Can you design reparameterizations that break the kernel limit (force feature learning, etc)?
3. Currently, the experiment in Figure 2 just reports a correlation between the alignment at initialization for various inits and the speed of training. If I understand correctly there is no reparameterization applied in this Figure. Is this plot surprising? I think it would be more interesting if the authors could **generate** better initializations based on their desired increased kernel alignment through reparameterization.

---

> ### Author Response · Authors · 2025-11-27
>
> Thank you for your thoughtful review. We are pleased that you found our paper interesting, with important contributions and good soundness, presentation, and theoretical value.
>
> **Regarding the weakness you have raised**
>
> > Few Example Applications of the Main Result...
>
> We believe that primarily theoretical papers, such as ours, are valuable even when they do not yield immediate practical implementations. In our view, a paper does not need to present a concrete method to be impactful. Providing an intuitive and rigorous explanation of an observed phenomenon can be just as important, as it can lay the groundwork for future practical advances.
>
> Nevertheless, we do allude to several possible direct practical directions that may emerge from our theoretical insights. One possibility is that now that we understand what quantity we truly maximize, we may be able to do so in a more direct way than through standard gradient descent. Another is the possibility of deliberately avoiding or promoting linearization, which we discuss in more detail in our response to your question on that topic.
>
> > The current empirical figure simply reports a correlation between alignment and training speed...
>
> We address this point in our response to your related question on the same issue.
>
> **Regarding your first questions:**
>
> > The authors claim that "maximizing the norm of the empirical loss gradient as defined in Equation 10, via symmetry transformations" provides a method to accelerate training. How is this done in practice? Are there analytical methods for this or is it done numerically?
>
> We maximize the norm of the empirical loss gradient as described in the work we aim to explain [1]. This is implemented numerically through an initial round of gradient descent over the parameters of the symmetry transformation, with the objective of maximizing the gradient-loss norm. This procedure is outlined in the introduction and in Appendix E. We thank you for your suggestion to clarify this core operation further. In response, we will include our implementation in the final version of the manuscript and emphasize this point more clearly in the introduction.
>
> > Can you use this theory to account for the inducement of lazy training through output rescaling ... Can you design reparameterizations that break the kernel limit (force feature learning, etc)?
>
> Yes we can. This point is discussed in the paper in lines 317-321 and in Appendix D.2, but it should have been addressed more explicitly and will be further emphasized. For linear symmetries, which constitute the bulk of relevant neural network symmetries, our results show that we cannot use such symmetries to disrupt the NTK limit in practical implementations. However, many practical networks that adhere to the NTK limit also possess nonlinear symmetries. As we show in Appendix D.1, these nonlinear symmetries can be implemented to avoid or even control convergence to the NTK limit. We thank you for highlighting the need to emphasize this impotent point, and we will ensure it is emphasized in the next version of our work.
>
> ---
>
> [1] Zhao, Bo, et al. "Improving Convergence and Generalization Using Parameter Symmetries." The Twelfth International Conference on Learning Representations.

---

### Official Review · Reviewer_DSih · 2025-10-31

**Soundness:** 2
**Presentation:** 2
**Contribution:** 3
**Rating:** 4
**Confidence:** 3

**Summary:**

This paper investigates the impact of parameter-space symmetries in neural networks on their learning behavior from the lens of neural tangent kernels (NTK). It is shown that transformations adhering to parameter-space symmetry preserve the NTK limit. Using the NTK perspective, it is revealed that the alignment between the NTK and data dictates the convergence rate of gradient descent. Limited experiments show the effectiveness of theoretical insights.

**Strengths:**

The novelty of this theoretical work lies in establishing the understanding of NTK under parameter-space symmetries and demonstrating that the alignment of NTK with data is a relevant metric for gradient descent convergence.

**Weaknesses:**

The paper has several weaknesses.

1. As a theoretical work, I found this paper to lack the necessary rigor for a reader to fully appreciate the theoretical results. In particular, Section 3.1 (Assumptions) and Theorem 4.1 could be improved with mathematical equations,
2. The impact of the theoretical results on the generalization of neural networks is unclear. In my opinion, this is a key weakness of the theory presented in this paper.
3. Experimental results are limited.

**Questions:**

See Weaknesses section.

---

> ### Author Response · Authors · 2025-11-27
>
> Thank you for your review, questions, and comments.
>
> > As a theoretical work, I found this paper to lack the necessary rigor for a reader to fully appreciate the theoretical results. In particular, Section 3.1 (Assumptions) and Theorem 4.1 could be improved with mathematical equations.
>
> All of the paper’s assumptions, statements, and proofs are written with full rigor. In Section 3.1, we present the assumptions in a condensed form for readability, and as noted at the end of the section, discuss them in details in Appendix A.
>
> We appreciate your suggestion that including more explicit equations could enhance clarity. We agree, and will revise the text to include additional mathematical expressions to improve readability. For instance, instead of just stating that our symmetries are “twice differentiable and well-defined around the parameter values at which they are applied,” we will also explicitly write that $g \in C^{2}(\mathcal{U}_\theta)$. Similarly, we will rewrite theorem 4.1 as the following:
>
> Given the setup of Lemma 3.1, after applying a parameter-space symmetry at an arbitrary training step $s'\in\mathbb{N}$ as described in Section 3.1, $\theta(s') \mapsto g(\theta(s'))$, the neural tangent kernel limit, as defined in Equation 7 with $m_n$ replacing $\frac{1}{\sqrt{n}}$, remains valid albeit with a different kernel:
> $$
> \left\Vert F(\theta(s))(x) - F^{lin}(s)(x) \right\Vert = O\left(m_n\right)\mapsto O\left(m_n\right) \ ,
> $$
>
> > The impact of the theoretical results on the generalization of neural networks is unclear. In my opinion, this is a key weakness of the theory presented in this paper.
>
> We believe that no single work can address an entire theoretical landscape, and that good research naturally raises follow-up questions. The focus of this paper is not the generalization properties of wide neural networks, but rather their optimization rate. In our empirical analyses, we have not observed any clear effect on generalization resulting from applying parameter-space symmetry operations intended to maximize gradient loss norm, consistent with the findings in [1], (aside from the expected improvement in convergence rate). We also see no theoretical reason to expect such effects. As noted in the discussion however, we agree that studying generalization in this context is a promising avenue for future research.
>
> > Experimental results are limited.
>
> This work is intended to provide a theoretical explanation for the empirical findings of prior studies such as [1], and therefore builds upon the empirical results presented there. As such, we considered it unnecessary to replicate the full set of experiments already conducted in those works.
>
> Nevertheless, we did include relevant empirical results that directly support each of our theoretical contributions. In particular, we illustrated the theorem in Figure 1 and Appendix E, and demonstrated the effectiveness of our new alignment measure in Figure 2 and Appendix F. We believe that for a theoretical work that rigorously establishes its claims through formal proofs, this level of empirical support is appropriate.
>
> We thank you again for your review, and we hope that our responses address your concerns regarding the overall quality and contribution of our work. Together with the revisions we plan to make based on your feedback and that of the other reviewers, we hope you will consider increasing your score.
>
> ---
>
> [1] Zhao, Bo, et al. "Improving Convergence and Generalization Using Parameter Symmetries." The Twelfth International Conference on Learning Representations.

---

### Author Response · Authors · 2025-11-28

Dear new area chair,

We understand that due to the leak on OpenReview, no discussion with the reviewers will take place, and they will not be able to update their scores. We are writing to clarify that although our average score is low, this score does not represent the current state of our paper, even before the rebuttals. This is because, as Reviewer 5haL (score: 2) noted, they had already reviewed our NeurIPS submission of the same work, where they supported acceptance, but a specific misunderstanding regarding the changes in this version led to their current score. We would prefer not to involve a review process from a previous conference, but since their current review is tied to that history, and as they can't participate anymore and change their score, we would like to give some necessary context.

The primary reason for the current score discrepancy stems from a specific misunderstanding regarding changes made between the NeurIPS submission of this work and the current submission. While we successfully addressed the feedback from the previous round, we also chose to streamline a specific derivation to improve clarity. Ironically, this simplification appears to have been interpreted as an omission by 5haL. Under normal circumstances, a brief clarification during the rebuttal phase would have resolved this concern and likely restored the reviewer's previous strong support. Since that dialogue is now impossible, we
would like to provide the necessary context.

In the NeurIPS review process, two reviewers gave mild support for acceptance, one opposed, and two others gave strong support, which we believe included this reviewer (based on the similarity between their current review and one of the NeurIPS reviews, and based on their explicit statement there that they supported acceptance). In the end, however, several concerns were raised:

1. Whether the efficacy of our new alignment metric was sufficiently proven.
2. Insufficient discussion of the nonlinear transformation case.
3. Concern about the effect of $C$ on the linearization, which we will soon expand on.

For the ICLR version, we addressed point (1) thoroughly (we proved the effectiveness of the new alignment metric in Section 5.2 and Appendix F and demonstrated it numerically in Figure 2). We also addressed point (2) in Section 3.2 and Appendix D.2, and expanded the discussion to clarify that almost all symmetry considered in practice for neural networks, are linear symmetries.

Regarding point (3), in the NeurIPS version of the paper, the effect of the transformation on the linearization of the network in the theorem was described not only as a function of $n$, but also as a function of $C$. There, we showed that after the symmetry transformation, the asymptotic bound becomes:
$$
O(m_n) \mapsto O(C^{4} m_n),
$$
where we use the two-variable big-$O$ asymptotic notation as a function of both $C$ and $n$.

While we explained in the paper (as in this version) that for any practical symmetry application, $C$ must be bounded in $n$, there was still a concern that even if we technically retain an $O\left(m_n\right)$ bound, the multiplicative factor of $C^{4}$ could obscure this behavior in practice. Unfortunately, this concern arose only near the end of the discussion period, so we failed to address it properly.

For the ICLR version, we initially included a clearer analysis of the effect of $C$, but we later felt that doing so created more confusion than clarity. As we explained to reviewer 5haL, the effect of choosing $1 < C$ is, in the worst case, equivalent to increasing the learning rate by a factor of $C$, since we can always absorb it into the learning rate $\eta$. Thus, this otherwise complicated discussion has no meaningful implications. Therefore, to avoid unnecessary complexity, we decided to analyze the convergence purely as a function of $n$.

We believe that when 5haL saw that we removed this discussion entirely, they felt that an important part of the previous version was missing, and this may have led to the score of 2. However, given our explanation of why $C$ cannot undermine the linearization beyond the effect of choosing a fine-tuned learning rate, and their strong support in the previous version of the paper even before the improvements we made for ICLR, we think they would have raised their score, possibly even to strong support, after reading our responses.

We would also like to point out that reviewer UXBr wrote a positive review, with only a few minor concerns that we have fully addressed, and scored us 3/4 in Soundness, Presentation, and Contribution. We believe there is a strong possibility that this reviewer too would have significantly increased their score.

Thank you very much for your time and consideration. We hope you will take all these factors into account.

---

### Author Response · Authors · 2025-12-03

Dear Area Chair,

Thank you for managing our paper through the recent transition. We have now addressed all key concerns raised by the reviewers, with revisions highlighted in blue for easy reference. We hope this updated version fully addresses the feedback.

---

### Meta-Review · Area_Chair_P5ve · 2026-01-05

**Summary:**

This paper claims to prove that the NTK limit is preserved by symmetry transformations. While I like the direction of studying how symmetry transformations change the NTK, I think there is a main problem with the theory. I agree with Reviewer 5haL that a lot of problems are hidden under C.

All the invariance is proved under the assumption that the symmetry transformation is bounded, but this is not true for linear symmetries.
Take scaling symmetry for example, the factor C is inversely proportional to the scale change $\lambda$ (assuming that there is a scaling invariance and the transformation is $\theta \to \theta/\lambda$), and if we simply choose $\lambda = n^{1/4}$, this means that Eq. (24) is no longer vanishing but O(1), and this shows why the theory is essentially circular. This argument also answers the question from UXBr, showing that feature learning is indeed achievable with symmetry transformations, contrary to what the authors claim. Of course, I might be wrong in this simple counterexample, but the authors are obliged to address this concern in detail before acceptance.

**Reviewer Concerns:**

I think the answers are not convincing enough.

**Reviewer Scores:**

This is difficult to answer, but I feel that there will not be a change.

---

### Decision · Program_Chairs · 2026-01-26

Reject